# Uncertainty Quantification in Linear Regression With Mismatched Data

## Abstract

The fundamental assumption in regression analysis that each response-predictor pair corresponds to the same observational unit is not always valid, especially with mismatched data. This paper presents a novel approach for uncertainty quantification in linear regression when data mismatch occurs. Using the generalized fiducial inference framework, we develop a method to generate fiducial samples for constructing confidence intervals and measuring uncertainty in key regression parameters. We establish the theoretical properties of our approach and demonstrate its practical effectiveness through empirical tests on both simulated and real datasets. To our knowledge, this is the first study to explore uncertainty quantification for mismatched data in linear regression.

## 1 Introduction

Linear regression and its numerous extensions serve as fundamental tools in statistics and machine learning. A significant challenge arises when the correspondence between predictors $\boldsymbol{X}$ and responses $y$ is not fully established. Specifically, while both predictors and responses are available as separate datasets, the precise matching between them may be partially or entirely unknown. This issue, often referred to as "permuted data" or "sparsely permuted data" when only a small fraction of pairs are mismatched, has garnered significant attention in recent works (e.g., Pananjady et al., 2017a; Hsu et al., 2017; Tsakiris & Peng, 2019; Slawski et al., 2020; Zhang & Li, 2020; 2023a;b; Azadkia & Balabdaoui, 2024).

Historically, this problem has been studied under the umbrella of the "broken sample problem," a term introduced in the 1970s (e.g., DeGroot et al., 1971; DeGroot & Goel, 1980). Early research focused primarily on parameter estimation, such as regression coefficients, rather than recovering the correspondence between predictors and responses. This line of investigation is closely related to record linkage and statistical analysis based on merged datasets (e.g., Lahiri & Larsen, 2005). These challenges frequently arise in real-world applications such as the U.S. Census Bureau, where multiple data sources are integrated to address complex questions. In these contexts, mismatches and ambiguities in record linkage—often caused by the absence of unique identifiers or errors in quasi-identifiers (e.g., names, addresses, or dates of birth)—can lead to selection bias and pervasive outliers. For example, linkage errors resulting from privacy-preserving measures, such as the removal of social security numbers, may contaminate statistical analyses and hinder accurate parameter estimation. Additionally, identifying matching pairs may sometimes be undesirable due to confidentiality concerns. Notable examples include linkage attacks that exposed sensitive medical histories and the partial de-anonymization of Netflix movie rankings. These cases underscore the dual importance of mitigating the impact of mismatches while preserving data confidentiality (Domingo-Ferrer & Muralidhar, 2016). Another real example of mismatched data is discussed in Millimet (2024). Data are linked across sources or over time, and the unit of observation changes. A crosswalk is used to convert to a common unit of observation. For example, convert county-level data to congressional district-level data.

In recent work, Pananjady et al. (2017b) established the statistical limits of exact and approximate permutation recovery as a function of the signal-to-noise ratio (SNR), defined as the ratio of signal energy to noise variance. It was also shown in Pananjady et al. (2017b) that least squares estimation of the permutation matrix is NP-hard in general. To address these computational challenges, Hsu et al. (2017) proposed a

polynomial-time approximation algorithm and derived lower bounds on the required SNR for approximate signal recovery in noisy scenarios; related results can be found in Abid et al. (2017) and Slawski & Ben-David (2019). Additionally, Slawski & Ben-David (2019) investigated both signal and permutation recovery in cases where only a small fraction of the rows in the sensing matrix are permuted. Furthermore, Chakraborty & Datta (2024) proposed a robust Bayesian framework for this setting and developed an efficient posterior sampling scheme.

While prior works have focused on recovering the correct permutation and estimating regression parameters, they have largely overlooked the crucial aspect of uncertainty quantification. In regression with mismatched data, uncertainty arises not only from noise in the observations but also from ambiguity in the data correspondence itself. Without a principled way to quantify this uncertainty, models may provide unreliable estimates, leading to overconfident conclusions and poor decision-making. Confidence and prediction intervals are essential for assessing the reliability of estimates and ensuring robust inference, particularly in real-world applications where mismatches are unavoidable. To address this gap, our study develops a systematic approach to uncertainty quantification.

## 1.1 Contributions

Our main contributions are summarized as follows:

- We propose the first formal framework for uncertainty quantification in linear regression with mismatched data by introducing a generalized fiducial density for permutations. This approach enables systematic probabilistic analysis of model parameters and includes theoretical guarantees, such as asymptotic consistency (see Section 4).

- We develop a practical algorithm for generating fiducial samples, which facilitates the construction of confidence and prediction intervals (see Section 3). Numerical experiments demonstrate the effectiveness and robustness of the proposed method across various settings. As shown below, fiducial samples play a similar role to posterior samples in Bayesian analysis.

- Our method addresses the limitations of relying on one single permutation estimate for inference. By considering multiple potential permutation candidates through the framework of generalized fiducial density, our method provides a more stable and reliable method for uncertainty quantification.

## 1.2 Problem Definition

Suppose the data consists of $\boldsymbol{x}_i, y_i$, for $i = 1, \cdots, n$, where $\boldsymbol{x}_i \in \mathbb{R}^p$ and $y_i \in \mathbb{R}$. Due to errors in the record linkage process, some $\boldsymbol{x}_i$ may be paired with a non-corresponding $y_i$ (Slawski & Ben-David, 2019). If the number of such mismatches is known to be at most $k$, then there exists an unknown permutation $\varphi$ on $\{1, \cdots, n\}$ that moves at most $k$ indices. Consequently, $(y_1, \boldsymbol{x}_{\varphi(1)}), \ldots, (y_n, \boldsymbol{x}_{\varphi(n)})$ are independent realizations from the classical linear regression model:

$$y = \boldsymbol{x}^T \boldsymbol{\beta}^* + e, \quad \text{where } e \sim N(0, \sigma^2), \quad \boldsymbol{x} \perp e. \tag{1}$$

The general mismatch setting also considers scenarios with missing matches or one-to-many matches, where multiple elements in $\boldsymbol{y}$ may correspond to the same element in $\boldsymbol{x}$ (Slawski et al., 2020). Our proposed method can naturally accommodate such cases.

Let $\boldsymbol{\Pi}^*$ and $\boldsymbol{\Pi}^{*T}$ represent the matrix form of $\varphi$ and its inverse, respectively. Define $\boldsymbol{X} = (\boldsymbol{x}_1, \cdots, \boldsymbol{x}_n)^T$, $\boldsymbol{y} = (y_1, \cdots, y_n)^T$, and $\boldsymbol{e} = (e_1, \cdots, e_n)^T$. The model equation can be expressed as:

$$\boldsymbol{y} = \boldsymbol{\Pi}^* \boldsymbol{X} \boldsymbol{\beta}^* + \boldsymbol{e}, \tag{2}$$

$$\iff \boldsymbol{\Pi}^{*T} \boldsymbol{y} = \boldsymbol{X} \boldsymbol{\beta}^* + \boldsymbol{\Pi}^{*T} \boldsymbol{e}, \tag{3}$$

where $e_i \overset{\text{i.i.d.}}{\sim} N(0, \sigma^2)$.

Also, we assume a Gaussian design:

$$\boldsymbol{x}_i \overset{\text{i.i.d.}}{\sim} N(0, \boldsymbol{I}_p), \quad i = 1, \cdots, n. \tag{4}$$

Our results extend to the case

$$\boldsymbol{x}_i \overset{\text{i.i.d.}}{\sim} N(0, \boldsymbol{\Sigma}), \quad i = 1, \cdots, n,$$

where $\boldsymbol{\Sigma}$ is a symmetric positive definite matrix, by redefining the regression parameter as $\boldsymbol{\Sigma}^{\frac{1}{2}}\boldsymbol{\beta}^*$.

A toy example is provided in Table 1 for illustration. The first two columns represent the true pairs of data, while the third and fourth columns show the observed data with mismatches caused by the unknown permutation. For example, the first observation is $(\boldsymbol{x}_2, y_1)$, while the true pair is $(\boldsymbol{x}_1, y_1)$. The last two columns detail the permutation $\varphi(i)$, and the corresponding permutation matrix is shown in (5).

Table 1: A toy example

| Truth | | Observation | | i | $\varphi(i)$ |
|---|---|---|---|---|---|
| $y_1$ | $\boldsymbol{x}_1$ | $y_1$ | $\boldsymbol{x}_2$ | 1 | 2 |
| $y_2$ | $\boldsymbol{x}_2$ | $y_2$ | $\boldsymbol{x}_3$ | 2 | 3 |
| $y_3$ | $\boldsymbol{x}_3$ | $y_3$ | $\boldsymbol{x}_4$ | 3 | 4 |
| $y_4$ | $\boldsymbol{x}_4$ | $y_4$ | $\boldsymbol{x}_5$ | 4 | 5 |
| $y_5$ | $\boldsymbol{x}_5$ | $y_5$ | $\boldsymbol{x}_1$ | 5 | 1 |

$$\boldsymbol{\Pi}^* = \begin{bmatrix} 0 & 0 & 0 & 0 & 1 \\ 1 & 0 & 0 & 0 & 0 \\ 0 & 1 & 0 & 0 & 0 \\ 0 & 0 & 1 & 0 & 0 \\ 0 & 0 & 0 & 1 & 0 \end{bmatrix} \tag{5}$$

### 1.3 Outline

The remainder of this paper is organized as follows. Section 2 introduces the concept of generalized fiducial inference. In Section 3, we present the proposed method for applying generalized fiducial inference to linear regression with mismatched data. The theoretical properties of this method are developed in Section 4. Simulation studies and real data applications are discussed in Sections 5 and 6, respectively. Finally, the conclusions are drawn in Section 7, and technical details are deferred to the appendix.

### 1.4 Notation

Let $|S|$ denote the cardinality of a set $S$, and range($\boldsymbol{A}$) denote the column space of a matrix $\boldsymbol{A}$. Write $\langle \boldsymbol{u}, \boldsymbol{v} \rangle$ as the inner product of the vectors $\boldsymbol{u}$ and $\boldsymbol{v}$, and $\mathbb{S}^{n-1}$ as the unit sphere in $\mathbb{R}^n$. Also, $\|\cdot\|_0$ denotes the $\ell_0$-"norm," i.e., the number of non-zero entries of a vector. Lastly, we make use of the usual Big-O notation in terms of $O$ and $o$.

## 2 Introduction to Generalized Fiducial Inference

The concept of fiducial inference, originally proposed by Fisher, was developed as an alternative to Bayes' theorem in situations where prior information is unavailable. Fisher introduced a "switching principle," akin to the maximum likelihood approach, to derive a prior directly from the observed data (Fisher, 1930). In recent years, there have been numerous efforts to expand and refine the principles of fiducial inference, leading to various modern extensions. The generalized fiducial inference (GFI) framework has emerged as a prominent modern adaptation, demonstrating its effectiveness in tackling contemporary challenges such as graphon estimation (Su et al., 2022) and multi-task learning (Wei & Lee, 2023).

The foundation of GFI is to model the relationship between the observed data $\boldsymbol{Y}$ and the unknown parameters $\boldsymbol{\theta}$ through:

$$\boldsymbol{Y} = G(\boldsymbol{\theta}, \boldsymbol{U}), \tag{6}$$

where $G(\cdot, \cdot)$ is a deterministic and known function, and $\boldsymbol{U}$ is a random variable with a fully specified distribution (e.g., i.i.d. $N(0,1)$). The key aspect of GFI is the application of the switching principle, which treats the roles of $\boldsymbol{Y}$ and $\boldsymbol{\theta}$ as interchangeable in the likelihood function: $\boldsymbol{Y}$ is treated as fixed, while $\boldsymbol{\theta}$ is regarded as random.

Assuming that $G(\cdot, \cdot)$ has a well-defined inverse mapping, we define:

$$Q_Y(\boldsymbol{u}) = \{\boldsymbol{\theta} : \boldsymbol{Y} = G(\boldsymbol{\theta}, \boldsymbol{u})\},$$

where $\boldsymbol{u}$ is a realization of $\boldsymbol{U}$. It is important to note that this inverse mapping may not always exist. In cases where no $\boldsymbol{\theta}$ satisfies the equation, the corresponding $\boldsymbol{u}$ is excluded from the sample space, and probabilities are renormalized. Conversely, if multiple solutions exist, a single $\boldsymbol{\theta}$ is selected at random from the set $\{\boldsymbol{\theta} : \boldsymbol{Y} = G(\boldsymbol{\theta}, \boldsymbol{u})\}$ Hannig & Lee (2009).

Given that the distribution of $\boldsymbol{U}$ is fully known, random samples $\tilde{\boldsymbol{u}}_1, \tilde{\boldsymbol{u}}_2, \dots$ can be drawn, and corresponding fiducial samples

$$\tilde{\boldsymbol{\theta}}_1 = Q_Y(\tilde{\boldsymbol{u}}_1), \ \tilde{\boldsymbol{\theta}}_2 = Q_Y(\tilde{\boldsymbol{u}}_2), \ \dots$$

can be generated. These fiducial samples, analogous to Bayesian posterior samples, allow for statistical inference, such as constructing confidence intervals for $\boldsymbol{\theta}$. Furthermore, the corresponding fiducial density $r(\boldsymbol{\theta}|\boldsymbol{Y})$, analogous to a Bayesian posterior density, can also be obtained.

While the above description of GFI appears conceptually straightforward, it may not be directly suitable for all scenarios. Under a continuous distribution of data and some differentiability assumptions Hannig et al. (2016), the fiducial density $r(\boldsymbol{\theta}|\boldsymbol{Y})$ can be expressed as:

$$r(\boldsymbol{\theta}|\boldsymbol{Y}) = \frac{f(\boldsymbol{Y}, \boldsymbol{\theta})J(\boldsymbol{Y}, \boldsymbol{\theta})}{\int_\Theta f(\boldsymbol{Y}, \boldsymbol{\theta}')J(\boldsymbol{Y}, \boldsymbol{\theta}')d\boldsymbol{\theta}'}, \tag{7}$$

where $f(\boldsymbol{Y}, \boldsymbol{\theta})$ is the likelihood function, and

$$J(\boldsymbol{Y}, \boldsymbol{\theta}) = D\left(\left.\frac{\partial G(\boldsymbol{\theta}, \boldsymbol{u})}{\partial \boldsymbol{\theta}}\right|_{\boldsymbol{u}=G^{-1}(\boldsymbol{Y}, \boldsymbol{\theta})}\right),$$

with $D(\boldsymbol{A}) = \det(\boldsymbol{A}^\top \boldsymbol{A})^{1/2}$.

However, equation (7) assumes a fixed model dimension, which does not fit in the current problem where the number of mismatches $k$ is unknown. For problems involving model classes $\mathcal{M}$, the marginal fiducial probability for a model $M \in \mathcal{M}$ is given by:

$$r(M) \propto e^{-q(M)} \int_{\Theta_M} f_M(\boldsymbol{Y}, \boldsymbol{\theta}_M)J_M(\boldsymbol{Y}, \boldsymbol{\theta}_M)d\boldsymbol{\theta}_M, \tag{8}$$

where $q(M)$ is a penalty term of the model $M$.

In general, GFI is a valuable tool for modern statistical inference. By leveraging fiducial samples and fiducial densities, it provides an intuitive framework for analyzing parameter uncertainty without relying on the subjective priors required in Bayesian methods. Furthermore, GFI shares connections with Bayesian methodologies, such as the resemblance of $J(\boldsymbol{Y}, \boldsymbol{\theta})$ to Jeffreys' prior, and its behavior is analogous to that of a Bayesian posterior.

## 3 GFI For Mismatched Data

This section applies the above GFI framework to the current problem of regression with mismatched data. The parameter set is

$$\boldsymbol{\theta}_M = \{\boldsymbol{\Pi}_M, \sigma_M^2, \boldsymbol{\beta}_M\},$$

where $M$ denotes a specific permutation. With these, the data generation function (6) is

$$\boldsymbol{Y} = G(\boldsymbol{\theta}_M, \boldsymbol{U}) = \boldsymbol{\Pi}_M \boldsymbol{X} \boldsymbol{\beta}_M + \sigma_M U, \tag{9}$$

where $U \sim N(0, \boldsymbol{I}_n)$. Let $\boldsymbol{P_X}$ and $\boldsymbol{P_X^\perp}$ denote the orthogonal projection onto the range of $\boldsymbol{X}$ and its orthogonal complement, respectively. Note that $\boldsymbol{P_X^\perp} = \boldsymbol{I}_n - \boldsymbol{P_X}$. Define the set of indices of mismatched pairs as $S^* = \{i : \boldsymbol{\Pi}_{ii}^* = 0\}$ and also define $S_M = \{i : \boldsymbol{\Pi}_{M,ii} = 0\}$, where $\boldsymbol{\Pi}_{M,ii}$ is the $(i,i)$-th element of the permutation matrix $\boldsymbol{\Pi}_M$.

Following the framework of GFI, we calculate:

$$J_M(\boldsymbol{y}, \boldsymbol{\theta}_M) = \sigma_M^{-1} |\det(\boldsymbol{X}^T \boldsymbol{X})|^{\frac{1}{2}} \|\boldsymbol{P_X^\perp} \boldsymbol{\Pi}_M^T \boldsymbol{y}\|_2,$$

where $\|\boldsymbol{P_X^\perp} \boldsymbol{\Pi}_M^T \boldsymbol{y}\|_2^2$ represents the residual sum of squares under the permutation $\boldsymbol{\Pi}_M$. The calculation of $J_M(\boldsymbol{y}, \boldsymbol{\theta}_M)$ is similar to Lai et al. (2015).

Next, the likelihood function is given by:

$$f_M(\boldsymbol{y}, \boldsymbol{\theta}_M) = (2\pi \sigma_M^2)^{-\frac{n}{2}} \exp \left\{ -\frac{1}{2\sigma_M^2} \|\boldsymbol{P_X^\perp} \boldsymbol{\Pi}_M^T \boldsymbol{y}\|_2^2 \right\}.$$

We incorporate the Bayesian Information Criterion (BIC) as the penalty, defined as

$$q(M) = \frac{1}{2} \log n \cdot |S_M|.$$

The previous studies within GFI shows the promising result in model selection using BIC (Lai et al., 2015; Hannig et al., 2016). Consequently, the generalized fiducial density of a candidate model $M$ is then:

$$r(\boldsymbol{\Pi}_M) \propto R(\boldsymbol{\Pi}_M) = \|\boldsymbol{P_X^\perp} \boldsymbol{\Pi}_M^T \boldsymbol{y}\|_2^{p+1-n} e^{-\frac{1}{2} \log n \cdot |S_M|}. \tag{10}$$

It is important to note that the estimation of $\sigma_M^2$ and $\boldsymbol{\beta}_M$ relies on $\boldsymbol{\Pi}_M$, which is the central component of the problem. Note that $r(\boldsymbol{\Pi}_M)$ includes the residual sum of squares $\|\boldsymbol{P_X^\perp} \boldsymbol{\Pi}_M^T \boldsymbol{y}\|_2^2$ and the penalty $-\frac{1}{2} \log n \cdot |S_M|$. Notice that once $\boldsymbol{\Pi}_M$ is specified, $\sigma_M^2$ and $\boldsymbol{\beta}_M$ can be uniquely estimated. Therefore, for simplicity, we refer $\boldsymbol{\Pi}_M$ as the parameter set instead of $\boldsymbol{\theta}_M$.

### 3.1 Practical Generation of Fiducial Sample

This section develops a practical procedure for generating fiducial samples. We begin by presenting a modified version of the algorithm in Slawski et al. (2020) for estimating $\boldsymbol{\Pi}^*$. The main steps of this algorithm are:

1. Obtain $\hat{\boldsymbol{\beta}}$ and $\hat{\boldsymbol{g}}$ as
$$\underset{\boldsymbol{\beta} \in \mathbb{R}^p, \boldsymbol{g} \in \mathbb{R}^n}{\arg \min} \frac{1}{n} \|\boldsymbol{y} - \boldsymbol{X}\boldsymbol{\beta} - \sqrt{n}\boldsymbol{g}\|_2^2 + \lambda \|\boldsymbol{g}\|_1,$$

   where $\lambda > 0$ is a tuning parameter, $\boldsymbol{g}$ targets $\boldsymbol{g}^* = \frac{1}{\sqrt{n}}(\boldsymbol{\Pi}^* - \boldsymbol{I}_n)\boldsymbol{X}\boldsymbol{\beta}^*$. Note that $\boldsymbol{g}^*$ signifies the locations of the mismatches; to be more specific $i \in S^* \iff g_i^* \neq 0$.

2. Detect the mismatches by inspecting the centered magnitudes of the $\hat{g}_i$'s in $\hat{\boldsymbol{g}} = (\hat{g}_1, \cdots, \hat{g}_n)^T$. The rationale behind this is that the larger the magnitude of $\hat{g}_i$, the more likely the $i$-th observation is an outlier, which also implies the more likely it is a mismatch. Define the set of outlier indices as:
$$\hat{S} = \{i : |\hat{g}_i - \text{median}(\hat{g}_i)| > l \cdot \text{MAD}\},$$

   where MAD is the median absolute deviation of $\hat{g}_i$ and $l$ is a tuning parameter. Notice that $k$ can be naturally estimated by
$$\hat{k} = |\hat{S}|. \tag{11}$$

3. Refine $\hat{\boldsymbol{\beta}}$ by performing ordinary least squares on the subset of data points $\{(\boldsymbol{x_i}, y_i) : i \notin \hat{S}\}$.

4. Estimate $\boldsymbol{\Pi}^*$ by

$$\min_{\boldsymbol{\Pi}} \|\boldsymbol{y} - \boldsymbol{\Pi}\boldsymbol{X}\hat{\boldsymbol{\beta}}\|_2^2$$

subject to for any $i \notin \hat{S}$, the row $i$ of $\boldsymbol{\Pi}$ and $\boldsymbol{I}_n$ are identical, i.e, only the rows in $\hat{S}$ are permuted.

Note that Step 4 is a linear assignment problem Burkard et al. (2012), a specific linear program that can be solved efficiently by the Hungarian Algorithm. As mentioned before, the purpose of Step 2 is to locate the indices of mismatches, which are then used as inputs for Steps 3 and 4. The rationale is that the index $i$ is mismatched if $|\hat{g}_i|$ is large. Step 2 is a novel and practical addition to the original algorithm proposed by Slawski et al. (2020). This original algorithm uses the true $k$ as a threshold in Step 2 to detect outliers. This algorithm is also included in simulations. Finally, we note that Step 2 can be replaced with alternative methods designed for outlier detection.

Next, we experiment with different tuning parameters $(\lambda, l)$ to generate a set of model candidates, denoted as $\widehat{\mathcal{P}} = \{\boldsymbol{\Pi}_M\}$. When the chosen tuning parameters are reasonably close to the optimal values, it is expected that $\sum_{\boldsymbol{\Pi}_M \in \widehat{\mathcal{P}}} r(\boldsymbol{\Pi}_M)$ is very close to 1.

Finally we can generate a fiducial sample $\{\tilde{\boldsymbol{\Pi}}_M, \tilde{\sigma}_M^2, \tilde{\boldsymbol{\beta}}_M\}$ with the following steps:

1. Generate a $\boldsymbol{\Pi}_M$ from

$$\hat{r}(\boldsymbol{\Pi}_M) = \frac{R(\boldsymbol{\Pi}_M)}{\sum_{\boldsymbol{\Pi}_M \in \widehat{\mathcal{P}}} R(\boldsymbol{\Pi}_M)},$$

where $R(\boldsymbol{\Pi}_M)$ is defined in (10).

2. With $\boldsymbol{\Pi}_M$, generate a $\sigma_M^2$ from

$$\sigma_M^2 \sim \|\boldsymbol{P}_{\boldsymbol{X}}^{\perp}\boldsymbol{\Pi}_M^T\boldsymbol{y}\|_2^2/\chi_{n-p}^2.$$

3. With $\boldsymbol{\Pi}_M, \sigma_M^2$, generate a $\boldsymbol{\beta}_M$ from

$$\boldsymbol{\beta}_M \sim N((\boldsymbol{X}^T\boldsymbol{X})^{-1}\boldsymbol{X}^T\boldsymbol{\Pi}_M^T\boldsymbol{y}, \sigma_M^2(\boldsymbol{X}^T\boldsymbol{X})^{-1}).$$

Step 2 and 3 are derived from the distributional assumptions on the residuals $e_i$'s.

### 3.2 Inference with Fiducial Sample

Repeating the above steps, one can obtain multiple copies of $\{(\tilde{\boldsymbol{\Pi}}_M, \tilde{\sigma}_M^2, \tilde{\boldsymbol{\beta}}_M)\}$. With these copies, one can then form point estimates and confidence intervals for the unknown parameters, which are similar to Bayesian posterior samples. The statistical properties can be found in Corollary 3.1 (Lai et al., 2015).

For any $\boldsymbol{x}_i$, the corresponding conditional mean is

$$\mu_{\boldsymbol{x}_i} = \mathbb{E}[y_i|\boldsymbol{x}_i] = \boldsymbol{x}_i^T\boldsymbol{\beta}.$$

Based on one copy of $\{(\tilde{\boldsymbol{\Pi}}_M, \tilde{\sigma}_M^2, \tilde{\boldsymbol{\beta}}_M)\}$, a fiducial sample $\tilde{\mu}_{\boldsymbol{x}_i}$ can be obtained as $\tilde{\mu}_{\boldsymbol{x}_i} = \boldsymbol{x}_i^T\tilde{\boldsymbol{\beta}}$.

Let $\hat{\mu}_{\boldsymbol{x}_i}$ be the sample mean and $\hat{S}_{\boldsymbol{x}_i}$ the sample standard deviation of these $\tilde{\mu}_{\boldsymbol{x}_i}$. Then, $\hat{\mu}_{\boldsymbol{x}_i}$ is a point estimate for $\mu_{\boldsymbol{x}_i}$. Additionally, one can construct a $100(1-\alpha)\%$ confidence interval for $\mu_{\boldsymbol{x}_i}$:

$$\hat{\mu}_{\boldsymbol{x}_i} \pm \hat{S}_{\boldsymbol{x}_i}Z_\alpha,$$

where $Z_\alpha$ is the upper $(100\alpha)$th percentile of standard normal distribution.

Similarly, a prediction point estimate and a prediction interval can be obtained for any new observation $y_i$ at $\boldsymbol{x}_i$. Compared to the conditional mean $\mu_{\boldsymbol{x}_i}$, a new observation $y_i$ exhibits greater variability. A fiducial sample can be obtained by adding a noise term: $\tilde{\mu}_{\boldsymbol{x}_i}^* = \tilde{\mu}_{\boldsymbol{x}_i} + \tilde{\sigma}_M z$ with $z \sim N(0, 1)$. Let $\hat{\mu}_{\boldsymbol{x}_i}^*$ be the sample mean and $\hat{S}_{\boldsymbol{x}_i}^*$ the sample standard deviation of all $\tilde{\mu}_{\boldsymbol{x}_i}$ values. Then $\hat{\mu}_{\boldsymbol{x}_i}^*$ is a point estimate for $y_i$. A $100(1-\alpha)\%$ prediction interval for $y_i$ can then be constructed as:

$$\hat{\mu}_{\boldsymbol{x}_i}^* \pm \hat{S}_{\boldsymbol{x}_i}^*Z_\alpha.$$

## 4  Theoretical Properties

This section establishes the asymptotic properties of the generalized fiducial-based method described above. We begin by introducing the necessary notations. Let $\mathcal{P}_n$ denote the set of all permutation matrices in $\mathbb{R}^{n \times n}$, and define

$$\mathcal{P}_{n,k} = \{\mathbf{\Pi}_M \in \mathcal{P}_n : |S_M| \leq k\},$$

where $|S_M|$ represents the number of mismatches associated with the permutation matrix $\Pi_M$.

We define the signal-to-noise ratio (SNR) as:

$$\text{SNR} = \frac{\|\boldsymbol{\beta}^*\|_2^2}{\sigma^2}.$$

The SNR serves as a critical assumption to establish the consistency of the proposed method.

For a compact and symmetric[1] set $S \subseteq \mathbb{R}^n$, its Gaussian width is defined as:

$$\omega(S) = \mathbb{E} \sup_{x \in S} |\langle g, x \rangle|, \quad g \sim N(0, \boldsymbol{I}_n), \tag{12}$$

where $\langle g, x \rangle$ represents the inner product of $g$ and $x$. The Gaussian width is a complexity measure widely used in high-dimensional linear inverse problems (Cai et al., 2016).

In our theoretical analysis, we consider the set $\mathcal{P}_{n,\tilde{k}}$ with $\tilde{k} \geq k$, which allows for at most $\tilde{k}$ mismatches. The Gaussian width is applied to the set

$$\mathcal{T} = \bigcup_{\mathbf{\Pi} \in \mathcal{P}_{n,\tilde{k}}} \{\text{range}(\mathbf{\Pi} - \mathbf{\Pi}^{*T})\} \cap \mathbb{S}^{n-1}. \tag{13}$$

Notice for any $\boldsymbol{v} \in \mathbb{R}^n, \mathbf{\Pi} \in \mathcal{P}_{n,\tilde{k}}$,

$$\|(\mathbf{\Pi} - \mathbf{\Pi}^{*T})v\|_0 \leq \|\mathbf{\Pi}v\|_0 + \|\mathbf{\Pi}^{*T}v\|_0 \leq 2\tilde{k}.$$

So $\mathcal{T} \subseteq B_0(2\tilde{k}, n) \cap \mathbb{S}^{n-1}$, where the set

$$B_0(r, n) = \{\boldsymbol{v} \in \mathbb{R}^n : \|\boldsymbol{v}\|_0 \leq r\}.$$

By Lemma 2.3 in Plan & Vershynin (2012), we have

$$\omega(\mathcal{T}) \leq \omega(B_0(2\tilde{k}, n) \cap \mathbb{S}^{n-1}) \leq 3.5\sqrt{2\tilde{k} \log \frac{en}{2\tilde{k}}}. \tag{14}$$

The above result is utilized for concentration inequalities in the proof. Now, we establish the key theoretical results for our method.

**Theorem 4.1.** *Under the model (3), assume $\lim_{n \to \infty} \frac{p}{n} < 1$. Also, assume for any constant $\delta \in (0, 1)$, there exist constants $\epsilon \in (0, 1)$ and $C_1 > 0$ such that*

$$\frac{\|\boldsymbol{\beta}^*\|_2^2}{\sigma^2} \geq 2\tilde{k} \left( \frac{3\tilde{k}nC_1}{\delta(n-p)(1-\epsilon)} \right)^2 \log \frac{n}{\tilde{k}} - 1 \quad and \quad \tilde{k} \log \frac{n}{\tilde{k}} = o(n).$$

*Then for any $\mathbf{\Pi}_M \in \mathcal{P}_{n,\tilde{k}}, \mathbf{\Pi}_M \neq \mathbf{\Pi}^*$, there exists a positive constant $C_2$ such that*

$$\log \frac{r(\mathbf{\Pi}_M)}{r(\mathbf{\Pi}^*)} \leq -C_2 \tilde{k} \log \frac{n}{\tilde{k}} \tag{15}$$

---

[1] A set $S$ is symmetric if $x \in S$ implies $-x \in S$.

*with probability at least*

$$1 - \delta - 2\exp\left\{-(n-p)(\frac{\epsilon^2}{4} - \frac{\epsilon^3}{6})\right\} - \exp\left\{\frac{-C_1^2(\tilde{k}\log\frac{n}{k})}{8}\right\}.$$

*As a consequence, $\frac{r(\mathbf{\Pi}_M)}{r(\mathbf{\Pi}^*)} \to 0$ as $n \to \infty$. Additionally, let $\mathcal{Q}_{n,l} = \{\mathbf{\Pi}_M \in \mathcal{P}_n : |S_M| = l\}, l = 0, 1, \cdots, $ then*

$$P\left(\sum_{\mathbf{\Pi}_M \in \bigcup_{l=1}^{\tilde{k}} \mathcal{Q}_{n,l}} \frac{r(\mathbf{\Pi}^*)}{r(\mathbf{\Pi}_M)} \to 1\right) \geq 1 - \delta, \; n \to \infty. \tag{16}$$

The proof can be found in Appendix B. The proof relies on analyzing the relative likelihood of different permutations. Specifically, the theorem shows that the ratio of fiducial densities $\frac{r(\mathbf{\Pi}^*)}{r(\mathbf{\Pi}_M)}$ diverges to infinity for any incorrect permutation $\mathbf{\Pi}_M$, implying that the probability assigned to incorrect permutations asymptotically vanishes. This result justifies the practical approach for generating model candidates described in Section 3.1.

To establish this result, we derive a lower bound on the difference between the residual sum of squares (RSS) under $\mathbf{\Pi}^*$ and that under any incorrect $\mathbf{\Pi}_M$.

Lastly, we remark that when compared to the theoretical results in Slawski & Ben-David (2019), our results are more general in the sense that the set $\mathcal{P}_{n,\tilde{k}}$ contains the true value of $k$. We believe our work better aligns with real-world scenarios.

## 5 Simulations

This section reports results from simulation experiments that compared the practical performance of our proposed method with that of competing methods.

The simulation settings are as follows. The structure of the permutation $\varphi$ is similar to the toy example given in Table 1, where $\varphi(i) = i + 1$ for $i = 1, \cdots, k-1$, and $\varphi(k) = 1$. The number of predictors is $p = 40$. The true coefficient vector is set to $\boldsymbol{\beta} = b \cdot \mathbf{1}_p$, and the noise level is $\sigma = 1$. Each simulation is replicated 500 times. We evaluate the method under SNR=$(20, 40, 80)$, the mismatch proportions $k/n = (0.02, 0.05, 0.08)$ and the sample size $n = (100, 200, 300)$. We note that the above SNR values are comparable to those in the simulations conducted in Slawski et al. (2020).

We consider the following five methods for comparison:

- oracle: Least-squares regression using true permutation matrix $\mathbf{\Pi}^*$ is first applied to estimate $\boldsymbol{\beta}^*$. Confidence and prediction intervals are then obtained using classical linear model results.

- naive: Least-squares regression is first applied directly to the original data for parameter estimation without accounting for mismatches. Then, classical linear model results are used to construct confidence and prediction intervals.

- proposed: The GFI-based method proposed in this paper.

- SBL-true$k$: With the true value of $k$, the method of Slawski et al. (2020) is applied to estimate $\boldsymbol{\beta}^*$ and $\mathbf{\Pi}^*$. Then, classical linear model results are used to obtain confidence and prediction intervals. Obviously, this method cannot be applied in practice, as the true value of $k$ is seldom known.

- SBL-BIC: Similar to SBL-true$k$ except the single model is selected by BIC from all model candidates in Section 3.1.

Notice that for most practical situations, the oracle and SBL-true$k$ methods cannot be applied, as the true values of $\mathbf{\Pi}^*$, $\boldsymbol{\beta}^*$, and $k$ are often unknown.

For the tuning parameters in the proposed method, the penalty parameter in the optimization is set as $\lambda = c_1 \frac{\hat{\sigma}}{\sqrt{n}}$, where $c_1 \in \{0.5, 1, 2, 4\}$, and $\hat{\sigma}$ is estimated using the MAD (median absolute deviation) of the residuals obtained from the robust regression with Huber loss.

To evaluate their relative performance, for each of the five methods, we construct the confidence intervals for the conditional expectation $E[y_i|\boldsymbol{x}_i]$ and the prediction intervals for $y_i$ at the levels 90%, 95%, and 99%. The test set is set to $\boldsymbol{X} = \boldsymbol{I}_p$ to analyze the regression coefficient. Then $E[y_i|\boldsymbol{x}_i] = \beta_i, i = 1, \cdots, p$. We calculate the average empirical coverage rates and the average lengths of the confidence and prediction intervals. The proposed method constructs these intervals as described in Section 3.2, while classical linear model theory is used to build the intervals for the other methods. Additionally, we report the average mean squared errors of the coefficients $\text{MSE}(\beta_i) = \frac{1}{p} \sum_{i=1}^{p} (\hat{\beta}_i - \beta_i)^2$, with the standard errors provided in parentheses. Results are presented in Tables 2 to 17. Additional simulation results are shown in Appendix C.

No single method consistently outperforms the others across all scenarios. However, when considering the empirical coverage rates of the confidence intervals for $E[y_i|\boldsymbol{x}_i]$ (Tables 2, 4, and 6) and the prediction intervals for $y_i$ (Tables 3, 5, and 7), the proposed method demonstrates performance comparable to the oracle method; it achieves the best coverage rates in most cases. Even in instances where it does not achieve the closest coverage rate, the proposed method's results remain consistently close to the optimal rates. Notice that the confidence intervals constructed by the naive method are excessively wide and over-conservative, which makes them ineffective for meaningful inference. Moreover, when examining the tables summarizing the MSE results (Table 17), the proposed method achieves the smallest MSE in the majority of cases, second only to the oracle method. Finally, we note that the performance of the proposed method is similar to that of SBL-true$k$. However, as mentioned before, SBL-true$k$ cannot be applied in most practical situations as $k$ is typically unknown.

Table 2: Empirical coverage rates and average lengths of confidence intervals of $\beta_i$ with $p = 40$, $n = 200$, and $k = 10$. Best results are bolded (other than oracle and SBL-true$k$ methods).

| SNR | Method | 90% | Length | 95% | Length | 99% | Length |
|---|---|---|---|---|---|---|---|
| 20 | oracle | 89.8 | 0.259 | 94.7 | 0.308 | 98.8 | 0.405 |
| | proposed | **85.1** | 0.255 | **91.3** | 0.304 | **97.1** | 0.400 |
| | naive | 88.7 | 0.437 | 94.0 | 0.521 | 98.6 | 0.684 |
| | SBL-true$k$ | 84.7 | 0.253 | 91.1 | 0.302 | 97.3 | 0.396 |
| | SBL-BIC | 73.5 | 0.217 | 80.7 | 0.259 | 89.8 | 0.340 |
| 40 | oracle | 89.8 | 0.259 | 94.7 | 0.308 | 98.8 | 0.405 |
| | proposed | **86.0** | 0.257 | **92.2** | 0.306 | **97.8** | 0.403 |
| | naive | 88.6 | 0.559 | 94.0 | 0.666 | 98.6 | 0.875 |
| | SBL-true$k$ | 85.8 | 0.256 | 91.9 | 0.306 | 97.7 | 0.402 |
| | SBL-BIC | 77.8 | 0.229 | 84.9 | 0.273 | 93.0 | 0.359 |
| 80 | oracle | 89.8 | 0.259 | 94.7 | 0.308 | 98.8 | 0.405 |
| | proposed | **86.9** | 0.258 | **92.5** | 0.308 | **98.0** | 0.404 |
| | naive | 88.4 | 0.744 | 94.0 | 0.887 | 98.6 | 1.166 |
| | SBL-true$k$ | 87.1 | 0.259 | 92.7 | 0.309 | 98.1 | 0.406 |
| | SBL-BIC | 80.9 | 0.237 | 87.6 | 0.283 | 95.0 | 0.372 |

## 6  Real Data Example

This section reports results from applying the above five methods to the El Niño data set obtainable from the UCI Machine Learning Repository.[2]

This data set contains oceanographic and surface meteorological readings collected from buoys distributed across the equatorial Pacific. After excluding the missing values, there are $93,935$ records with the following

---

[2]https://archive.ics.uci.edu/dataset/122/el+nino

Table 3: Empirical coverage rates and average lengths of prediction intervals of $y_i$ with $p = 40$, $n = 200$, and $k = 10$. Best results are bolded (other than oracle and SBL-true$k$ methods).

| SNR | Method | 90% | Length | 95% | Length | 99% | Length |
|-----|--------|-----|--------|-----|--------|-----|--------|
| 20 | oracle | 89.9 | 3.298 | 94.9 | 3.936 | 98.9 | 5.196 |
| | proposed | **88.2** | 3.173 | **93.6** | 3.781 | **98.3** | 4.969 |
| | naive | 98.9 | 5.570 | 99.7 | 6.649 | 100.0 | 8.776 |
| | SBL-true$k$ | 89.1 | 3.228 | 94.3 | 3.853 | 98.7 | 5.087 |
| | SBL-BIC | 82.0 | 2.772 | 88.3 | 3.308 | 95.3 | 4.367 |
| 40 | oracle | 89.9 | 3.298 | 94.9 | 3.936 | 98.9 | 5.196 |
| | proposed | **88.9** | 3.206 | **94.0** | 3.821 | **98.5** | 5.021 |
| | naive | 99.8 | 7.124 | 100.0 | 8.504 | 100.0 | 11.226 |
| | SBL-true$k$ | 89.5 | 3.270 | 94.6 | 3.903 | 98.8 | 5.152 |
| | SBL-BIC | 84.7 | 2.921 | 90.9 | 3.487 | 96.9 | 4.603 |
| 80 | oracle | 89.9 | 3.298 | 94.9 | 3.936 | 98.9 | 5.196 |
| | proposed | **89.0** | 3.235 | **94.3** | 3.854 | **98.7** | 5.066 |
| | naive | 100.0 | 9.490 | 100.0 | 11.328 | 100.0 | 14.953 |
| | SBL-true$k$ | 89.8 | 3.305 | 94.9 | 3.946 | 98.9 | 5.208 |
| | SBL-BIC | 86.2 | 3.028 | 92.2 | 3.615 | 97.8 | 4.771 |

Table 4: Empirical coverage rates and average lengths of confidence intervals of $\beta_i$ with $p = 40$, $n = 200$, and $b = 1$. Best results are bolded (other than oracle and SBL-true$k$ methods).

| $k/n$ | Method | 90% | Length | 95% | Length | 99% | Length |
|-------|--------|-----|--------|-----|--------|-----|--------|
| 0.02 | oracle | 89.8 | 0.259 | 94.7 | 0.308 | 98.8 | 0.405 |
| | proposed | **87.4** | 0.256 | **93.0** | 0.305 | **98.0** | 0.401 |
| | naive | 89.3 | 0.403 | 94.4 | 0.480 | 98.8 | 0.630 |
| | SBL-true$k$ | 88.9 | 0.261 | 94.0 | 0.311 | 98.6 | 0.409 |
| | SBL-BIC | 80.4 | 0.233 | 87.1 | 0.278 | 94.4 | 0.365 |
| 0.05 | oracle | 89.8 | 0.259 | 94.7 | 0.308 | 98.8 | 0.405 |
| | proposed | **86.0** | 0.257 | **92.2** | 0.306 | **97.8** | 0.403 |
| | naive | 88.6 | 0.559 | 94.0 | 0.666 | 98.6 | 0.875 |
| | SBL-true$k$ | 85.8 | 0.256 | 91.9 | 0.306 | 97.7 | 0.402 |
| | SBL-BIC | 77.8 | 0.229 | 84.9 | 0.273 | 93.0 | 0.359 |
| 0.08 | oracle | 89.8 | 0.259 | 94.7 | 0.308 | 98.8 | 0.405 |
| | proposed | **84.1** | 0.256 | **90.6** | 0.305 | **96.8** | 0.400 |
| | naive | 87.4 | 0.679 | 93.0 | 0.809 | 98.4 | 1.064 |
| | SBL-true$k$ | 83.0 | 0.251 | 89.9 | 0.299 | 96.8 | 0.393 |
| | SBL-BIC | 74.1 | 0.223 | 81.3 | 0.265 | 90.6 | 0.348 |

Table 5: Empirical coverage rates and average lengths of prediction intervals of $y_i$ with $p = 40$, $n = 200$, and $b = 1$. Best results are bolded (other than oracle and SBL-true$k$ methods).

| $k/n$ | Method | 90% | Length | 95% | Length | 99% | Length |
|-------|--------|-----|--------|-----|--------|-----|--------|
| 0.02 | oracle | 89.9 | 3.298 | 94.9 | 3.936 | 98.9 | 5.196 |
| | proposed | **88.8** | 3.202 | **94.0** | 3.816 | **98.4** | 5.014 |
| | naive | 97.6 | 5.134 | 99.1 | 6.129 | 99.9 | 8.090 |
| | SBL-true$k$ | 90.3 | 3.333 | 95.1 | 3.979 | 99.0 | 5.252 |
| | SBL-BIC | 85.4 | 2.976 | 91.5 | 3.552 | 97.3 | 4.689 |
| 0.05 | oracle | 89.9 | 3.298 | 94.9 | 3.936 | 98.9 | 5.196 |
| | proposed | **88.9** | 3.206 | **94.0** | 3.821 | **98.5** | 5.021 |
| | naive | 99.8 | 7.124 | 100.0 | 8.504 | 100.0 | 11.226 |
| | SBL-true$k$ | 89.5 | 3.270 | 94.6 | 3.903 | 98.8 | 5.152 |
| | SBL-BIC | 84.7 | 2.921 | 90.9 | 3.487 | 96.9 | 4.603 |
| 0.08 | oracle | 89.9 | 3.298 | 94.9 | 3.936 | 98.9 | 5.196 |
| | proposed | **88.2** | 3.177 | **93.7** | 3.786 | **98.3** | 4.976 |
| | naive | 100.0 | 8.662 | 100.0 | 10.339 | 100.0 | 13.648 |
| | SBL-true$k$ | 88.7 | 3.199 | 94.0 | 3.819 | 98.6 | 5.041 |
| | SBL-BIC | 83.2 | 2.838 | 89.7 | 3.388 | 96.3 | 4.472 |

Table 6: Empirical coverage rates and average lengths of confidence intervals of $\beta_i$ with $p = 40$, $k = 10$, and $b = 1$. Best results are bolded (other than oracle and SBL-true$k$ methods).

| $n$ | Method | 90% | Length | 95% | Length | 99% | Length |
|---|---|---|---|---|---|---|---|
| 100 | oracle | 88.0 | 0.417 | 93.6 | 0.497 | 98.5 | 0.653 |
| | proposed | **81.2** | 0.434 | **87.9** | 0.517 | **95.0** | 0.679 |
| | naive | 87.5 | 1.200 | 93.1 | 1.430 | 98.2 | 1.879 |
| | SBL-true$k$ | 79.7 | 0.419 | 86.9 | 0.499 | 94.9 | 0.656 |
| | SBL-BIC | 61.8 | 0.315 | 68.9 | 0.375 | 80.0 | 0.493 |
| 200 | oracle | 89.8 | 0.259 | 94.7 | 0.308 | 98.8 | 0.405 |
| | proposed | **86.0** | 0.257 | **92.2** | 0.306 | **97.8** | 0.403 |
| | naive | 88.6 | 0.559 | 94.0 | 0.666 | 98.6 | 0.875 |
| | SBL-true$k$ | 85.8 | 0.256 | 91.9 | 0.306 | 97.7 | 0.402 |
| | SBL-BIC | 77.8 | 0.229 | 84.9 | 0.273 | 93.0 | 0.359 |
| 300 | oracle | 89.9 | 0.204 | 94.9 | 0.243 | 98.9 | 0.319 |
| | proposed | **87.2** | 0.201 | **93.0** | 0.240 | **98.2** | 0.315 |
| | naive | 88.7 | 0.383 | 94.0 | 0.457 | 98.7 | 0.601 |
| | SBL-true$k$ | 87.8 | 0.203 | 93.5 | 0.242 | 98.4 | 0.318 |
| | SBL-BIC | 82.8 | 0.190 | 89.6 | 0.227 | 96.5 | 0.298 |

Table 7: Empirical coverage rates and average lengths of prediction intervals of $y_i$ with $p = 40$, $k = 10$, and $b = 1$. Best results are bolded (other than oracle and SBL-true$k$ methods).

| $n$ | Method | 90% | Length | 95% | Length | 99% | Length |
|---|---|---|---|---|---|---|---|
| 100 | oracle | 89.3 | 3.298 | 94.5 | 3.949 | 98.9 | 5.252 |
| | proposed | **88.3** | 3.271 | **93.5** | 3.898 | **98.0** | 5.123 |
| | naive | 99.9 | 9.499 | 100.0 | 11.373 | 100.0 | 15.125 |
| | SBL-true$k$ | 89.2 | 3.315 | 94.4 | 3.969 | 98.8 | 5.279 |
| | SBL-BIC | 75.7 | 2.489 | 83.0 | 2.980 | 92.0 | 3.964 |
| 200 | oracle | 89.9 | 3.298 | 94.9 | 3.936 | 98.9 | 5.196 |
| | proposed | **88.9** | 3.206 | **94.0** | 3.821 | **98.5** | 5.021 |
| | naive | 99.8 | 7.124 | 100.0 | 8.504 | 100.0 | 11.226 |
| | SBL-true$k$ | 89.5 | 3.270 | 94.6 | 3.903 | 98.8 | 5.152 |
| | SBL-BIC | 84.7 | 2.921 | 90.9 | 3.487 | 96.9 | 4.603 |
| 300 | oracle | 89.9 | 3.296 | 95.0 | 3.932 | 99.1 | 5.181 |
| | proposed | **89.1** | 3.208 | **94.2** | 3.823 | **98.8** | 5.024 |
| | naive | 99.4 | 6.208 | 99.9 | 7.405 | 100.0 | 9.758 |
| | SBL-true$k$ | 89.9 | 3.287 | 94.9 | 3.921 | 99.0 | 5.167 |
| | SBL-BIC | 87.4 | 3.083 | 93.1 | 3.678 | 98.3 | 4.846 |

Table 8: Means and standard errors (in parentheses) of MSEs ($\times 10^{-3}$) for $\beta_i$. The smallest value for each experimental configuration is bolded, excluding those from the oracle method.

| | $n = 200, p = 40, k/n = 0.05$ | | |
|---|---|---|---|
| SNR | 20 | 40 | 80 |
| oracle | 6.296 (0.070) | 6.296 (0.070) | 6.296 (0.070) |
| proposed | **7.743** (0.100) | **7.469** (0.096) | **7.261** (0.088) |
| naive | 19.969 (0.418) | 33.469 (0.798) | 60.411 (1.558) |
| SBL-true$k$ | 7.925 (0.105) | 7.586 (0.096) | 7.336 (0.091) |
| SBL-BIC | 9.298 (0.142) | 8.662 (0.133) | 8.210 (0.117) |
| | $n = 200, p = 40, b = 1$ | | |
| $k/n$ | 0.02 | 0.05 | 0.08 |
| oracle | 6.296 (0.070) | 6.296 (0.070) | 6.296 (0.070) |
| proposed | 6.984 (0.089) | **7.469** (0.096) | **8.083** (0.107) |
| naive | 16.642 (0.440) | 33.469 (0.798) | 51.620 (1.078) |
| SBL-true$k$ | **6.781** (0.082) | 7.586 (0.096) | 8.325 (0.109) |
| SBL-BIC | 7.941 (0.117) | 8.662 (0.133) | 9.552 (0.146) |
| | $p = 40, b = 1, k = 10$ | | |
| $n$ | 100 | 200 | 300 |
| oracle | 17.480 (0.241) | 6.296 (0.070) | 3.885 (0.043) |
| proposed | **26.988** (0.544) | **7.469** (0.096) | **4.376** (0.051) |
| naive | 160.068 (4.001) | 33.469 (0.798) | 15.498 (0.345) |
| SBL-true$k$ | 27.566 (0.518) | 7.586 (0.096) | 4.380 (0.052) |
| SBL-BIC | 32.747 (0.610) | 8.662 (0.133) | 4.782 (0.061) |

attributes: ID, date, location (latitude and longitude), zonal and meridional wind speeds (*zon*, *mer*), relative humidity (*humidity*), air temperature (*air temp*), sea surface temperature, and subsurface temperatures down to a depth of 500 meters (*ss temp*).

For our analysis, we focused on a linear regression model with *air temp* as the response variable while using *zon*, *mer*, *humidity*, and *ss temp* as covariates (i.e., $p = 4$). To avoid the situation where $n \gg p$, we randomly drew $n = 300$ data points from the full dataset (of size $93{,}935$) and standardized the covariates. Then 80% of these 300 data points were designated as the training set while the remaining 20% were treated as the testing set. The response variable of the first 8% of the training set was randomly shuffled, resulting in a mismatching rate of 8%. The above five methods were first applied to the training set to fit the regression model, and then the fitted models were used to predict the responses in the testing set, as well as to obtain the corresponding prediction intervals. Denote the $i$-th predicted response in the testing set as $\hat{y}_i$. The process was repeated 500 times.

The results from the five methods are summarized in Table 9. In addition to the empirical coverage rates and lengths of the prediction intervals, we also report the averages and standard errors of the mean squared prediction error, $\text{MSPE}(y_i) = \frac{1}{\tilde{n}} \sum_{i=1}^{\tilde{n}} (\hat{y}_i - y_i)^2$, where $\tilde{n}$ is the sample size of the testing set (i.e., $\tilde{n} = 0.2 \times n = 60$). One can see that the proposed method consistently delivered high coverage rates close to the nominal levels, which demonstrates its reliability in capturing the true uncertainty of the air temperature prediction. Furthermore, apart from the oracle method, the proposed method attains the smallest MSPE, reinforcing its effectiveness. These findings agree with the empirical conclusions drawn from the simulation experiments.

Table 9: Empirical coverage rates and average lengths of prediction intervals of $y_i$ on real data. Also shown are the means and standard errors (in parentheses) of MSPE. Best results are bolded (other than oracle and SBL-true$k$ methods).

| Method | 90% | Length | 95% | Length | 99% | Length | MSPE |
|--------|-----|--------|-----|--------|-----|--------|------|
| oracle | 90.4 | 1.684 | 94.4 | 2.008 | 98.2 | 2.646 | 0.2642 (0.0029) |
| proposed | **88.1** | 1.569 | **92.5** | 1.870 | **97.1** | 2.457 | **0.2661** (0.0030) |
| naive | 97.1 | 2.577 | 98.6 | 3.074 | 99.7 | 4.050 | 0.2855 (0.0034) |
| SBL-true$k$ | 88.2 | 1.566 | 92.7 | 1.868 | 97.3 | 2.460 | 0.2663 (0.0030) |
| SBL-BIC | 87.5 | 1.553 | 92.2 | 1.852 | 96.9 | 2.440 | 0.2665 (0.0030) |

## 7 Conclusion

In this paper, we study linear regression problems where the correspondence between responses and predictors is lost. While recent work has extensively studied the statistical limits of permutation recovery and coefficient estimation, the important aspect of uncertainty quantification has remained unexplored. To bridge this gap, we proposed a novel method for uncertainty quantification under the framework of generalized fiducial inference. Specifically, we derived the generalized fiducial density for the problem and developed a practical method to generate fiducial samples from it to construct confidence and prediction intervals for key quantities of interest. Theoretical properties of the proposed method were established to ensure consistency. Through extensive simulations and real data applications, our method demonstrated performance comparable to the oracle method and outperformed other approaches.

One direction for future work is to extend the proposed method to handle multivariate response variables. While, in the context of permutation and coefficient estimation, extensions to multivariate settings have been explored in previous literature, adapting our method for uncertainty quantification may present new challenges, as we expect alternative proof techniques and methodologies to be required. Another direction for future work involves extending the method beyond classical linear models, such as generalized linear models. This extension would enhance flexibility in addressing categorical responses and other non-continuous data types and, therefore, broaden the proposed approach's applicability to more complex and diverse datasets.

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

## A  Lemmas

**Lemma A.1.** *For any positive $\delta, \gamma$ and $\mathbf{\Pi}_M \in \mathcal{P}_{n,\tilde{k}}, \mathbf{\Pi}_M \neq \mathbf{\Pi}^*$, if $\|\boldsymbol{\beta}^*\|_2^2 + \sigma^2 \geq \frac{2\tilde{k}^2\gamma^2}{\delta^2}$, then*

$$\mathbb{P}(\|(\mathbf{\Pi}_M^T - \mathbf{\Pi}^{*T})\boldsymbol{y}\|_2^2 \leq \gamma^2) \leq \delta. \tag{17}$$

*Proof.* Notice that only when $i \in S^* \cup S_M$, the $i$-th element of $(\mathbf{\Pi}_M^T - \mathbf{\Pi}^{*T})\boldsymbol{y}$ is nonzero. Also,

$$|S^* \cup S_M| \leq |S^*| + |S_M| \leq 2\tilde{k}.$$

We can rewrite the vector $(\mathbf{\Pi}_M^T - \mathbf{\Pi}^{*T})\boldsymbol{y} = (y_{\varphi_M^{-1}(1)} - y_{\varphi^{-1}(1)}, \cdots, y_{\varphi_M^{-1}(n)} - y_{\varphi^{-1}(n)})^T$,

$$\begin{aligned}
\mathbb{P}(\|(\mathbf{\Pi}_M^T - \mathbf{\Pi}^{*T})\boldsymbol{y}\|_2^2 \leq \gamma^2) &= \mathbb{P}(\sum_{i \in S^* \cup S_M} (y_{\varphi_M^{-1}(i)} - y_{\varphi^{-1}(i)})^2) \leq \gamma^2) \\
&\leq \sum_{i \in S^* \cup S_M} \mathbb{P}((y_{\varphi_M^{-1}(i)} - y_{\varphi^{-1}(i)})^2) \leq \gamma^2) \\
&\leq 2\tilde{k} \max_{i \neq j} \mathbb{P}((y_i - y_j)^2 \leq \gamma^2) \\
&\leq 2\tilde{k} \frac{\gamma}{\sqrt{2(\|\boldsymbol{\beta}^*\|_2^2 + \sigma^2)}}.
\end{aligned}$$

The last inequity comes from

$$y_i - y_j \sim N(0, 2(\|\boldsymbol{\beta}^*\|_2^2 + \sigma^2))$$

and if $g \sim N(0, 1)$,

$$\mathbb{P}(|g| < t) \leq \sqrt{\frac{2}{\pi}} t < t.$$

Finally, choose $2\tilde{k} \frac{\gamma}{\sqrt{2(\|\boldsymbol{\beta}^*\|_2^2 + \sigma^2)}} \leq \delta$. $\qquad\square$

**Lemma A.2.** *(Lemma 2.2 in Dasgupta & Gupta (2003)) Let $\boldsymbol{P}$ denote the orthogonal projection on an $N$-dimensional subspace of $\mathbb{R}^p$ chosen uniformly at random from the Grassmannian $G(p, N)$. Then for any $\boldsymbol{v} \in \mathbb{R}^p$ and any $\epsilon \in (0, 1)$,*

$$\mathbb{P}((1 - \epsilon)\|\boldsymbol{v}\|_2^2 \leq \frac{p}{N}\|\boldsymbol{P}\boldsymbol{v}\| \leq (1 + \epsilon)\|\boldsymbol{v}\|_2^2) \geq 1 - 2\exp\{-N(\frac{\epsilon^2}{4} - \frac{\epsilon^3}{6})\} \tag{18}$$

**Lemma A.3.** *(Concentration of Gaussian processes (Boucheron et al., 2013)) Let $S$ be a closed subset of the unit sphere in $\mathbb{R}^n$ with Gaussian width $\omega(S)$, and let $g \sim N(0, \boldsymbol{I}_n)$. Then for any positive $t$,*

$$\mathbb{P}(\sup_{x \in S} |\langle g, x \rangle| \geq \omega(S) + t) \leq e^{-\frac{t^2}{2}}. \tag{19}$$

## B  Proof of Theorem 4.1

*Proof.* **Proof of the first section** (15): At first, we rewrite the fraction,

$$\log \frac{r(\mathbf{\Pi}^*)}{r(\mathbf{\Pi}_M)} = \frac{n - p - 1}{2} \log(\frac{\|\boldsymbol{P}_{\boldsymbol{X}}^{\perp} \mathbf{\Pi}_M^T \boldsymbol{y}\|_2^2}{\|\boldsymbol{P}_{\boldsymbol{X}}^{\perp} \mathbf{\Pi}^{*T} \boldsymbol{y}\|_2^2}) - \frac{k - |S_M|}{2} \log n.$$

Next we will find the lower bound for $T_1 = \|\boldsymbol{P}_{\boldsymbol{X}}^{\perp} \mathbf{\Pi}_M^T \boldsymbol{y}\|_2^2 - \|\boldsymbol{P}_{\boldsymbol{X}}^{\perp} \mathbf{\Pi}^{*T} \boldsymbol{y}\|_2^2$.

$$\begin{aligned}
T_1 &= \|\boldsymbol{P}_{\boldsymbol{X}}^{\perp} \mathbf{\Pi}^{*T} \boldsymbol{y} + \boldsymbol{P}_{\boldsymbol{X}}^{\perp} (\mathbf{\Pi}_M^T - \mathbf{\Pi}^{*T})\boldsymbol{y}\|_2^2 - \|\boldsymbol{P}_{\boldsymbol{X}}^{\perp} \mathbf{\Pi}^{*T} \boldsymbol{y}\|_2^2 \\
&= \|\boldsymbol{P}_{\boldsymbol{X}}^{\perp} (\mathbf{\Pi}_M^T - \mathbf{\Pi}^{*T})\boldsymbol{y}\|_2^2 + 2\langle \boldsymbol{P}_{\boldsymbol{X}}^{\perp} \mathbf{\Pi}^{*T} \boldsymbol{y}, (\mathbf{\Pi}_M^T - \mathbf{\Pi}^{*T})\boldsymbol{y} \rangle \\
&= \|\boldsymbol{P}_{\boldsymbol{X}}^{\perp} (\mathbf{\Pi}_M^T - \mathbf{\Pi}^{*T})\boldsymbol{y}\|_2^2 + 2\langle \boldsymbol{P}_{\boldsymbol{X}}^{\perp} \boldsymbol{e}, (\mathbf{\Pi}_M^T - \mathbf{\Pi}^{*T})\boldsymbol{y} \rangle.
\end{aligned} \tag{20}$$

Let $\boldsymbol{v}_M = (\boldsymbol{\Pi}_M^T - \boldsymbol{\Pi}^{*T})\boldsymbol{y}$ and apply lemma A.2 to the first term $\|\boldsymbol{P}_{\boldsymbol{X}}^{\perp}(\boldsymbol{\Pi}_M^T - \boldsymbol{\Pi}^{*T})\boldsymbol{y}\|_2^2$ with $P = P_{\boldsymbol{X}}^{\perp}$, i.e.

$$\mathbb{P}(\frac{n}{n-p}\|\boldsymbol{P}_{\boldsymbol{X}}^{\perp}\boldsymbol{v}_M\| \geq (1-\epsilon)\|\boldsymbol{v}_M\|_2^2) \geq 1 - 2\exp\{-(n-p)(\epsilon^2/4 - \epsilon^3/6)\}. \tag{21}$$

As for the second term $\langle \boldsymbol{P}_{\boldsymbol{X}}^{\perp}\boldsymbol{e}, \boldsymbol{v}_M\rangle$, let $t_1 > 0$,

$$t_2 = t_1 - \mathbb{E}[\sup_{\boldsymbol{v}\in\mathcal{T}}|\langle \boldsymbol{P}_{\boldsymbol{X}}^{\perp}\boldsymbol{e}, \boldsymbol{v}\rangle\|\boldsymbol{P}_{\boldsymbol{X}}^{\perp}] \geq t_1 - \sigma\omega(\mathcal{T}),$$

we have

$$\begin{aligned}
\mathbb{P}(|\langle \boldsymbol{P}_{\boldsymbol{X}}^{\perp}\boldsymbol{e}, \frac{\boldsymbol{v}_M}{\|\boldsymbol{v}_M\|_2}\rangle| > t_1|\boldsymbol{P}_{\boldsymbol{X}}^{\perp}) &\leq \mathbb{P}(\sup_{\boldsymbol{v}\in\mathcal{T}}|\langle \boldsymbol{P}_{\boldsymbol{X}}^{\perp}\boldsymbol{e}, \boldsymbol{v}\rangle| > t_1|\boldsymbol{P}_{\boldsymbol{X}}^{\perp}) \\
&\leq \mathbb{P}(\sup_{\boldsymbol{v}\in\mathcal{T}}|\langle \boldsymbol{P}_{\boldsymbol{X}}^{\perp}\boldsymbol{e}, \boldsymbol{v}\rangle| > \mathbb{E}[\sup_{\boldsymbol{v}\in\mathcal{T}}|\langle \boldsymbol{P}_{\boldsymbol{X}}^{\perp}\boldsymbol{e}, \boldsymbol{v}\rangle\|\boldsymbol{P}_{\boldsymbol{X}}^{\perp}] + t_2|\boldsymbol{P}_{\boldsymbol{X}}^{\perp}) \\
&\leq \mathbb{P}(\sup_{\boldsymbol{v}\in\mathcal{T}}\langle \boldsymbol{P}_{\boldsymbol{X}}^{\perp}\boldsymbol{e}, \boldsymbol{v}\rangle > t_2 + \sigma\omega(\mathcal{T})|\boldsymbol{P}_{\boldsymbol{X}}^{\perp}) \\
&\leq e^{-\frac{t_2^2}{2\sigma^2}}. \tag{22}
\end{aligned}$$

The third inequality a consequence of the Sudakov-Fernique comparison inequality (Adler & Taylor, 2009, Theorem 2.2.3) and the last inequality comes from Lemma A.3. Choose the constant $C_1^2 \geq 99$ such that

$$C_1\sqrt{\tilde{k}\log\frac{n}{\tilde{k}}} > 7\sqrt{2\tilde{k}\log\frac{en}{2\tilde{k}}} \geq 2\omega(\mathcal{T})$$

according to (14). Because for large $n$,

$$\frac{C_1\sqrt{\tilde{k}\log\frac{n}{\tilde{k}}}}{7\sqrt{2\tilde{k}\log\frac{en}{2\tilde{k}}}} = \sqrt{\frac{C_1^2\log\frac{n}{\tilde{k}}}{98(\log\frac{n}{\tilde{k}} + \log\frac{e}{2})}} > 1.$$

Then let $t_1 = C_1\sigma\sqrt{\tilde{k}\log\frac{n}{\tilde{k}}}$ and $t_2 \geq t_1 - \sigma\omega(\mathcal{T}) \geq \frac{C_1\sigma}{2}\sqrt{\tilde{k}\log\frac{n}{\tilde{k}}}$ in inequality (22), we have

$$\mathbb{P}(|\langle \boldsymbol{P}_{\boldsymbol{X}}^{\perp}\boldsymbol{e}, \frac{\boldsymbol{v}_M}{\|\boldsymbol{v}_M\|_2}\rangle| > C_1\sigma\sqrt{\tilde{k}\log\frac{n}{\tilde{k}}}|\boldsymbol{P}_{\boldsymbol{X}}^{\perp}) \leq \exp\{-\frac{C_1^2\tilde{k}\log\frac{n}{\tilde{k}}}{8}\}$$

$$\mathbb{P}(\langle \boldsymbol{P}_{\boldsymbol{X}}^{\perp}\boldsymbol{e}, \frac{\boldsymbol{v}_M}{\|\boldsymbol{v}_M\|_2}\rangle > -C_1\sigma\sqrt{\tilde{k}\log\frac{n}{\tilde{k}}}|\boldsymbol{P}_{\boldsymbol{X}}^{\perp}) \geq 1 - \exp\{-\frac{C_1^2\tilde{k}\log\frac{n}{\tilde{k}}}{8}\}.$$

Combine the inequality of the first term and the second term of $T_1$ in equation (20),

$$T_1 \geq \|\boldsymbol{v}_M\|_2(\frac{n-p}{n}(1-\epsilon)\|\boldsymbol{v}_M\|_2 - 2\sigma C_1\sqrt{\tilde{k}\log\frac{n}{\tilde{k}}})$$

with probability at least

$$1 - 2\exp\{-(n-p)(\frac{\epsilon^2}{4} - \frac{\epsilon^3}{6})\} - \exp\{-\frac{C_1^2(\tilde{k}\log\frac{n}{\tilde{k}})}{8}\}.$$

Next we use (17) with $\frac{n-p}{n}(1-\epsilon)\gamma \geq 3\sigma C_1\sqrt{\tilde{k}\log\frac{n}{\tilde{k}}}$ and the new assumption of SNR,

$$T_1 \geq \gamma(\frac{n-p}{n}(1-\epsilon)\gamma - 2\sigma C_1\sqrt{\tilde{k}\log\frac{n}{\tilde{k}}}) \geq C_3\sigma^2\tilde{k}\log\frac{n}{\tilde{k}},$$

where $C_3 = \frac{3C_1^2 n}{(n-p)(1-\epsilon)}$. Next notice that $\|\boldsymbol{P}_{\boldsymbol{X}}^{\perp}\boldsymbol{\Pi}^{*T}\boldsymbol{y}\|_2^2 = \|\boldsymbol{P}_{\boldsymbol{X}}^{\perp}e\|_2^2 \sim \sigma^2\chi_{n-p}^2$ since $\mathrm{rank}(\boldsymbol{P}_{\boldsymbol{X}}^{\perp}) = n - p$. By Chebyshev's inequality, we have

$$\mathbb{P}(|\chi_{n-p}^2 - (n-p)| > \log n\sqrt{2(n-p)}) \le \frac{1}{(\log n)^2}.$$

Finally, we look at the target fraction, for sufficiently large n,

$$\begin{aligned}
\log\frac{r(\boldsymbol{\Pi}_M)}{r(\boldsymbol{\Pi}^*)} &= -\frac{n-p-1}{2}\log(1 + \frac{T_1}{\|\boldsymbol{P}_{\boldsymbol{X}}^{\perp}\boldsymbol{\Pi}^{*T}\boldsymbol{y}\|_2^2}) + \frac{1}{2}\log n(k - |S_M|) \\
&\le -\frac{n-p-1}{2}\log(1 + \frac{C_3\tilde{k}\log\frac{n}{\tilde{k}}}{n-p+\log n\sqrt{2(n-p)}}) + \frac{k-|S_M|}{2}\log n \\
&\le -\frac{1}{2}C_3\tilde{k}\log\frac{n}{\tilde{k}} + \frac{k-|S_M|}{2}\log n,
\end{aligned}$$

where $C_3 = \frac{3C_1^2 n}{(n-p)(1-\epsilon)} > 3C_1^2 \ge 3 \times 99$.

Then there exists a positive constant $C_2$ such that $\log\frac{r(\boldsymbol{\Pi}_M)}{r(\boldsymbol{\Pi}^*)} \le -C_2\tilde{k}\log\frac{n}{k}$.

**Proof of the second section** (16):

Note that $|\mathcal{Q}_{n,l}| \le n^l$, we have

$$\begin{aligned}
\sum_{\boldsymbol{\Pi}_M\in\bigcup_{l=1}^{\tilde{k}}\mathcal{Q}_{n,l},\boldsymbol{\Pi}_M\neq\boldsymbol{\Pi}^*}\frac{r(\boldsymbol{\Pi}_M)}{r(\boldsymbol{\Pi}^*)} &\le \sum_{l=0}^{\tilde{k}}|\mathcal{Q}_{n,l}|\max_{\boldsymbol{\Pi}_M\in\mathcal{Q}_{n,l},\boldsymbol{\Pi}_M\neq\boldsymbol{\Pi}^*}\frac{r(\boldsymbol{\Pi}_M)}{r(\boldsymbol{\Pi}^*)} \\
&\le \sum_{l=0}^{\tilde{k}}n^l\exp\{-\frac{1}{2}C_3\tilde{k}\log\frac{n}{\tilde{k}} + \frac{k-l}{2}\log n\} \\
&= \exp\{-\frac{1}{2}C_3\tilde{k}\log\frac{n}{\tilde{k}} + \frac{k}{2}\log n\}\sum_{l=0}^{\tilde{k}}n^{\frac{l}{2}} \\
&= \frac{1-n^{\frac{\tilde{k}+1}{2}}}{1-n^{\frac{1}{2}}}\exp\{-\frac{1}{2}C_3\tilde{k}\log\frac{n}{\tilde{k}} + \frac{k}{2}\log n\} \\
&\le \frac{3}{2}\exp\{\frac{\tilde{k}}{2}\log n - \frac{1}{2}C_3\tilde{k}\log\frac{n}{\tilde{k}} + \frac{k}{2}\log n\} \xrightarrow{\mathbb{P}} 0.
\end{aligned}$$

The above is equivalent to

$$\sum_{\boldsymbol{\Pi}_M\in\bigcup_{l=1}^{\tilde{k}}\mathcal{Q}_{n,l}}\frac{r(\boldsymbol{\Pi}^*)}{r(\boldsymbol{\Pi}_M)} \xrightarrow{\mathbb{P}} 1.$$

$\square$

## C   Additional Simulations

We add the requested simulation results here.

Table 10: Empirical coverage rates and average lengths of confidence intervals of $\beta_i$ with $p = 40$, $n = 200$, and $k = 10$. Best results are bolded (other than oracle and SBL-true$k$ methods).

| SNR | Method | 90% | Length | 95% | Length | 99% | Length |
|-----|--------|------|--------|------|--------|------|--------|
| 5 | oracle | 89.8 | 0.259 | 94.7 | 0.308 | 98.8 | 0.405 |
|   | proposed | 82.6 | 0.251 | 89.3 | 0.300 | 96.0 | 0.394 |
|   | naive | **88.9** | 0.314 | **94.0** | 0.374 | **98.6** | 0.492 |
|   | SBL-true$k$ | 80.7 | 0.244 | 87.9 | 0.291 | 95.7 | 0.383 |
|   | SBL-BIC | 66.1 | 0.200 | 73.9 | 0.238 | 84.2 | 0.312 |
| 10 | oracle | 89.8 | 0.259 | 94.7 | 0.308 | 98.8 | 0.405 |
|   | proposed | 83.9 | 0.252 | 90.1 | 0.300 | 96.5 | 0.394 |
|   | naive | **88.9** | 0.360 | **94.1** | 0.429 | **98.6** | 0.564 |
|   | SBL-true$k$ | 83.1 | 0.249 | 89.7 | 0.296 | 96.4 | 0.390 |
|   | SBL-BIC | 69.9 | 0.208 | 77.2 | 0.248 | 87.1 | 0.326 |
| 15 | oracle | 89.8 | 0.259 | 94.7 | 0.308 | 98.8 | 0.405 |
|   | proposed | 84.4 | 0.253 | 90.5 | 0.301 | 96.7 | 0.396 |
|   | naive | **88.7** | 0.400 | **93.9** | 0.477 | **98.6** | 0.627 |
|   | SBL-true$k$ | 84.0 | 0.252 | 90.6 | 0.300 | 97.0 | 0.394 |
|   | SBL-BIC | 72.9 | 0.216 | 80.1 | 0.257 | 89.4 | 0.338 |

Table 11: Empirical coverage rates and average lengths of prediction intervals of $y_i$ with $p = 40$, $n = 200$, and $k = 10$. Best results are bolded (other than oracle and SBL-true$k$ methods).

| SNR | Method | 90% | Length | 95% | Length | 99% | Length |
|-----|--------|------|--------|------|--------|------|--------|
| 5 | oracle | 89.9 | 3.298 | 94.9 | 3.936 | 98.9 | 5.196 |
|   | proposed | **87.1** | 3.104 | **92.7** | 3.699 | **97.9** | 4.861 |
|   | naive | 94.9 | 4.005 | 97.9 | 4.780 | 99.8 | 6.310 |
|   | SBL-true$k$ | 87.8 | 3.117 | 93.3 | 3.721 | 98.3 | 4.912 |
|   | SBL-BIC | 77.4 | 2.545 | 84.0 | 3.037 | 92.1 | 4.009 |
| 10 | oracle | 89.9 | 3.298 | 94.9 | 3.936 | 98.9 | 5.196 |
|   | proposed | **87.5** | 3.123 | **93.1** | 3.722 | **98.0** | 4.891 |
|   | naive | 97.1 | 4.590 | 99.0 | 5.480 | 99.9 | 7.233 |
|   | SBL-true$k$ | 88.4 | 3.173 | 93.8 | 3.787 | 98.6 | 4.999 |
|   | SBL-BIC | 79.6 | 2.657 | 86.1 | 3.172 | 93.8 | 4.187 |
| 15 | oracle | 89.9 | 3.298 | 94.9 | 3.936 | 98.9 | 5.196 |
|   | proposed | **87.8** | 3.140 | **93.3** | 3.741 | **98.1** | 4.917 |
|   | naive | 98.2 | 5.105 | 99.5 | 6.094 | 100.0 | 8.044 |
|   | SBL-true$k$ | 88.9 | 3.208 | 94.1 | 3.829 | 98.7 | 5.054 |
|   | SBL-BIC | 81.7 | 2.755 | 88.0 | 3.289 | 95.2 | 4.341 |

Table 12: Means and standard errors (in parentheses) of MSE ($\times 10^{-3}$) for $\beta_i$. The smallest value for each experimental configuration is bolded, excluding those from the oracle method.

| | $p = 40, n = 200, k = 10$ | | |
|---|---|---|---|
| SNR | 5 | 10 | 15 |
| oracle | 6.296 (0.070) | 6.296 (0.070) | 6.296 (0.070) |
| proposed | **8.407** (0.120) | **8.072** (0.106) | **7.941** (0.105) |
| naive | 9.798 (0.141) | 13.198 (0.231) | 16.587 (0.324) |
| SBL-true$k$ | 8.849 (0.118) | 8.374 (0.114) | 8.072 (0.107) |
| SBL-BIC | 10.560 (0.155) | 9.936 (0.154) | 9.455 (0.143) |

Table 13: Empirical coverage rates and average lengths of confidence intervals of $\beta_i$ with $p = 40$, $n = 200$, and $b = 2$. Best results are bolded (other than oracle and SBL-true$k$ methods).

| $k/n$ | Method | 90% | Length | 95% | Length | 99% | Length |
|---|---|---|---|---|---|---|---|
| 0.1 | oracle | 89.8 | 0.259 | 94.7 | 0.308 | 98.8 | 0.405 |
| | proposed | **85.4** | 0.257 | **91.4** | 0.307 | **97.3** | 0.403 |
| | naive | 86.2 | 1.419 | 92.5 | 1.691 | 98.2 | 2.222 |
| | SBL-true$k$ | 85.4 | 0.256 | 91.5 | 0.305 | 97.4 | 0.401 |
| | SBL-BIC | 80.5 | 0.240 | 87.5 | 0.286 | 95.0 | 0.376 |
| 0.12 | oracle | 89.8 | 0.259 | 94.7 | 0.308 | 98.8 | 0.405 |
| | proposed | **84.2** | 0.256 | **90.6** | 0.305 | **97.0** | 0.401 |
| | naive | 85.8 | 1.549 | 92.2 | 1.846 | 98.0 | 2.426 |
| | SBL-true$k$ | 84.0 | 0.254 | 90.3 | 0.303 | 96.9 | 0.398 |
| | SBL-BIC | 79.0 | 0.237 | 86.0 | 0.282 | 94.2 | 0.371 |
| 0.14 | oracle | 89.8 | 0.259 | 94.7 | 0.308 | 98.8 | 0.405 |
| | proposed | **82.8** | 0.254 | **89.3** | 0.303 | **96.3** | 0.398 |
| | naive | 85.1 | 1.664 | 91.8 | 1.983 | 97.8 | 2.607 |
| | SBL-true$k$ | 82.5 | 0.252 | 89.1 | 0.300 | 96.2 | 0.394 |
| | SBL-BIC | 77.1 | 0.234 | 84.3 | 0.279 | 93.1 | 0.367 |

Table 14: Empirical coverage rates and average lengths of prediction intervals of $y_i$ with $p = 40$, $n = 200$, and $b = 2$. Best results are bolded (other than oracle and SBL-true$k$ methods).

| $k/n$ | Method | 90% | Length | 95% | Length | 99% | Length |
|---|---|---|---|---|---|---|---|
| 0.1 | oracle | 89.9 | 3.298 | 94.9 | 3.936 | 98.9 | 5.196 |
| | proposed | **88.8** | 3.215 | **94.1** | 3.831 | **98.5** | 5.035 |
| | naive | 100.0 | 18.093 | 100.0 | 21.597 | 100.0 | 28.509 |
| | SBL-true$k$ | 89.6 | 3.267 | 94.5 | 3.900 | 98.8 | 5.148 |
| | SBL-BIC | 86.8 | 3.059 | 92.6 | 3.652 | 97.9 | 4.821 |
| 0.12 | oracle | 89.9 | 3.298 | 94.9 | 3.936 | 98.9 | 5.196 |
| | proposed | **88.5** | 3.188 | **93.8** | 3.799 | **98.3** | 4.993 |
| | naive | 100.0 | 19.751 | 100.0 | 23.577 | 100.0 | 31.122 |
| | SBL-true$k$ | 89.2 | 3.240 | 94.4 | 3.868 | 98.8 | 5.106 |
| | SBL-BIC | 86.5 | 3.021 | 92.4 | 3.606 | 97.7 | 4.760 |
| 0.14 | oracle | 89.9 | 3.298 | 94.9 | 3.936 | 98.9 | 5.196 |
| | proposed | **88.1** | 3.159 | **93.5** | 3.764 | **98.3** | 4.947 |
| | naive | 100.0 | 21.223 | 100.0 | 25.333 | 100.0 | 33.441 |
| | SBL-true$k$ | 88.9 | 3.210 | 94.1 | 3.832 | 98.6 | 5.058 |
| | SBL-BIC | 85.8 | 2.985 | 91.9 | 3.563 | 97.6 | 4.704 |

Table 15: Empirical coverage rates and average lengths of confidence intervals of $\beta_i$ with $n = 200$, $SNR = 40$, and $k = 10$. Best results are bolded (other than oracle and SBL-true$k$ methods).

| $p$ | Method | 90% | Length | 95% | Length | 99% | Length |
|-----|--------|-----|--------|-----|--------|-----|--------|
| 10 | oracle | 89.7 | 0.238 | 94.6 | 0.284 | 98.8 | 0.373 |
| | proposed | **86.7** | 0.235 | **92.8** | 0.280 | **97.9** | 0.368 |
| | naive | 83.2 | 0.525 | 89.8 | 0.626 | 96.9 | 0.823 |
| | SBL-true$k$ | 87.4 | 0.237 | 93.0 | 0.282 | 98.2 | 0.371 |
| | SBL-BIC | 84.6 | 0.228 | 90.7 | 0.271 | 96.9 | 0.357 |
| 20 | oracle | 89.4 | 0.244 | 94.5 | 0.291 | 98.9 | 0.383 |
| | proposed | **86.1** | 0.241 | **92.0** | 0.287 | **97.7** | 0.378 |
| | naive | 86.9 | 0.531 | 92.8 | 0.633 | 98.3 | 0.832 |
| | SBL-true$k$ | 86.6 | 0.243 | 92.3 | 0.289 | 98.0 | 0.380 |
| | SBL-BIC | 82.9 | 0.230 | 89.2 | 0.274 | 96.1 | 0.360 |
| 40 | oracle | 89.8 | 0.259 | 94.7 | 0.308 | 98.8 | 0.405 |
| | proposed | **86.0** | 0.257 | **92.2** | 0.306 | **97.8** | 0.403 |
| | naive | 88.6 | 0.559 | 94.0 | 0.666 | 98.6 | 0.875 |
| | SBL-true$k$ | 85.8 | 0.256 | 91.9 | 0.306 | 97.7 | 0.402 |
| | SBL-BIC | 77.8 | 0.229 | 84.9 | 0.273 | 93.0 | 0.359 |
| 60 | oracle | 89.6 | 0.276 | 94.6 | 0.329 | 98.9 | 0.433 |
| | proposed | **85.1** | 0.275 | **91.2** | 0.328 | **97.4** | 0.431 |
| | naive | 88.6 | 0.600 | 94.0 | 0.715 | 98.6 | 0.939 |
| | SBL-true$k$ | 85.2 | 0.275 | 91.5 | 0.327 | 97.6 | 0.430 |
| | SBL-BIC | 65.9 | 0.209 | 73.3 | 0.249 | 83.8 | 0.327 |
| 80 | oracle | 89.4 | 0.299 | 94.5 | 0.356 | 98.8 | 0.468 |
| | proposed | **84.9** | 0.299 | **90.9** | 0.357 | **97.1** | 0.469 |
| | naive | 88.8 | 0.656 | 94.1 | 0.782 | 98.6 | 1.028 |
| | SBL-true$k$ | 83.9 | 0.296 | 90.3 | 0.353 | 97.0 | 0.464 |
| | SBL-BIC | 55.8 | 0.190 | 63.0 | 0.227 | 74.5 | 0.298 |

Table 16: Empirical coverage rates and average lengths of prediction intervals of $y_i$ with $n = 200$, $SNR = 40$, and $k = 10$. Best results are bolded (other than oracle and SBL-true$k$ methods).

| $p$ | Method | 90% | Length | 95% | Length | 99% | Length |
|---|---|---|---|---|---|---|---|
| 10 | oracle | 90.1 | 3.307 | 95.2 | 3.947 | 99.1 | 5.206 |
| | proposed | **88.8** | 3.204 | **94.2** | 3.817 | **98.8** | 5.017 |
| | naive | 99.7 | 7.294 | 99.9 | 8.704 | 100.0 | 11.482 |
| | SBL-true$k$ | 89.9 | 3.285 | 95.1 | 3.920 | 99.0 | 5.171 |
| | SBL-BIC | 88.1 | 3.161 | 93.9 | 3.772 | 98.6 | 4.975 |
| 20 | oracle | 89.8 | 3.298 | 94.8 | 3.937 | 98.9 | 5.194 |
| | proposed | **88.5** | 3.190 | **93.7** | 3.801 | **98.5** | 4.996 |
| | naive | 99.7 | 7.176 | 99.9 | 8.564 | 100.0 | 11.299 |
| | SBL-true$k$ | 89.5 | 3.278 | 94.7 | 3.912 | 98.9 | 5.162 |
| | SBL-BIC | 87.2 | 3.103 | 92.9 | 3.703 | 98.2 | 4.886 |
| 40 | oracle | 89.9 | 3.298 | 94.9 | 3.936 | 98.9 | 5.196 |
| | proposed | **88.9** | 3.206 | **94.0** | 3.821 | **98.5** | 5.021 |
| | naive | 99.8 | 7.124 | 100.0 | 8.504 | 100.0 | 11.226 |
| | SBL-true$k$ | 89.5 | 3.270 | 94.6 | 3.903 | 98.8 | 5.152 |
| | SBL-BIC | 84.7 | 2.921 | 90.9 | 3.487 | 96.9 | 4.603 |
| 60 | oracle | 89.8 | 3.295 | 94.8 | 3.934 | 99.0 | 5.197 |
| | proposed | **88.5** | 3.209 | **93.9** | 3.824 | **98.5** | 5.025 |
| | naive | 99.6 | 7.156 | 99.9 | 8.544 | 100.0 | 11.286 |
| | SBL-true$k$ | 89.5 | 3.275 | 94.6 | 3.911 | 98.9 | 5.166 |
| | SBL-BIC | 76.3 | 2.492 | 83.6 | 2.976 | 92.3 | 3.930 |
| 80 | oracle | 89.6 | 3.310 | 94.8 | 3.954 | 99.0 | 5.227 |
| | proposed | **88.6** | 3.234 | **93.9** | 3.854 | **98.5** | 5.065 |
| | naive | 99.7 | 7.264 | 99.9 | 8.676 | 100.0 | 11.469 |
| | SBL-true$k$ | 89.2 | 3.278 | 94.4 | 3.915 | 98.9 | 5.176 |
| | SBL-BIC | 68.0 | 2.104 | 75.5 | 2.513 | 85.9 | 3.322 |

Table 17: Means and standard errors (in parentheses) of MSE ($\times 10^{-3}$) for $\beta_i$. The smallest value for each experimental configuration is bolded, excluding those from the oracle method.

| | $n = 200, p = 40, b = 2$ | | |
|---|---|---|---|
| $k/n$ | 0.1 | 0.12 | 0.14 |
| oracle | 6.296 (0.070) | 6.296 (0.070) | 6.296 (0.070) |
| proposed | **7.816** (0.101) | **8.186** (0.108) | **8.725** (0.116) |
| naive | 233.899 (4.804) | 287.586 (5.596) | 343.136 (6.358) |
| SBL-true$k$ | 7.869 (0.105) | 8.300 (0.111) | 8.753 (0.118) |
| SBL-BIC | 8.448 (0.115) | 8.978 (0.127) | 9.618 (0.134) |
| | $p = 40, SNR = 40, k = 10$ | | |
| $p$ | 10 | 20 | 40 |
| oracle | 5.316 (0.106) | 5.767 (0.082) | 6.296 (0.070) |
| proposed | 6.068 (0.122) | 6.624 (0.104) | **7.469** (0.096) |
| naive | 39.606 (1.385) | 33.281 (0.945) | 33.469 (0.798) |
| SBL-true$k$ | **6.012** (0.124) | **6.624** (0.097) | 7.586 (0.096) |
| SBL-BIC | 6.371 (0.131) | 7.039 (0.116) | 8.662 (0.133) |
| $p$ | 60 | 80 | |
| oracle | 7.222 (0.072) | 8.518 (0.078) | |
| proposed | **8.906** (0.104) | **10.846** (0.119) | |
| naive | 37.983 (0.885) | 44.756 (0.987) | |
| SBL-true$k$ | 8.980 (0.098) | 11.198 (0.131) | |
| SBL-BIC | 11.771 (0.180) | 14.853 (0.182) | |

In the following simulation, assume $\boldsymbol{x}_i \sim N(0, \boldsymbol{\Sigma}), \boldsymbol{\Sigma}_{i,j} = \rho_X^{|i-j|}$.

Table 18: Empirical coverage rates and average lengths of confidence intervals of $\beta_i$ with $n = 200$, $p = 40$, $SNR = 40$, and $k = 10$. Best results are bolded (other than oracle and SBL-true$k$ methods).

| $\rho_X$ | Method | 90% | Length | 95% | Length | 99% | Length |
|---|---|---|---|---|---|---|---|
| 0.2 | oracle | 89.9 | 0.269 | 94.7 | 0.320 | 98.8 | 0.421 |
| | proposed | **86.6** | 0.268 | **92.4** | 0.319 | **97.9** | 0.419 |
| | naive | 88.8 | 0.682 | 94.2 | 0.812 | 98.7 | 1.068 |
| | SBL-true$k$ | 86.7 | 0.268 | 92.4 | 0.320 | 98.0 | 0.420 |
| | SBL-BIC | 80.0 | 0.244 | 86.6 | 0.291 | 94.2 | 0.382 |
| 0.4 | oracle | 90.1 | 0.303 | 94.8 | 0.361 | 98.8 | 0.474 |
| | proposed | **87.6** | 0.302 | **93.0** | 0.360 | **98.1** | 0.473 |
| | naive | 89.2 | 0.923 | 94.3 | 1.100 | 98.9 | 1.446 |
| | SBL-true$k$ | 87.7 | 0.304 | 93.1 | 0.362 | 98.2 | 0.476 |
| | SBL-BIC | 82.0 | 0.280 | 88.4 | 0.333 | 95.3 | 0.438 |
| 0.6 | oracle | 90.0 | 0.374 | 94.9 | 0.446 | 98.9 | 0.586 |
| | proposed | **88.3** | 0.375 | **93.5** | 0.446 | **98.3** | 0.587 |
| | naive | 89.3 | 1.442 | 94.5 | 1.718 | 98.9 | 2.258 |
| | SBL-true$k$ | 88.3 | 0.376 | 93.8 | 0.448 | 98.3 | 0.589 |
| | SBL-BIC | 84.3 | 0.355 | 90.4 | 0.423 | 96.8 | 0.556 |

Table 19: Empirical coverage rates and average lengths of prediction intervals of $y_i$ with $n = 200$, $p = 40$, $SNR = 40$, and $k = 10$. Best results are bolded (other than oracle and SBL-true$k$ methods).

| $\rho_X$ | Method | 90% | Length | 95% | Length | 99% | Length |
|---|---|---|---|---|---|---|---|
| 0.2 | oracle | 89.9 | 3.298 | 94.9 | 3.937 | 98.9 | 5.197 |
| | proposed | **88.8** | 3.216 | **94.2** | 3.833 | **98.6** | 5.037 |
| | naive | 99.9 | 8.361 | 100.0 | 9.981 | 100.0 | 13.175 |
| | SBL-true$k$ | 89.7 | 3.291 | 94.7 | 3.929 | 98.9 | 5.186 |
| | SBL-BIC | 85.7 | 2.995 | 91.9 | 3.575 | 97.5 | 4.719 |
| 0.4 | oracle | 89.9 | 3.301 | 94.8 | 3.941 | 99.0 | 5.202 |
| | proposed | **89.0** | 3.234 | **94.2** | 3.854 | **98.6** | 5.065 |
| | naive | 100.0 | 10.067 | 100.0 | 12.017 | 100.0 | 15.863 |
| | SBL-true$k$ | 89.9 | 3.311 | 94.8 | 3.953 | 98.9 | 5.217 |
| | SBL-BIC | 86.6 | 3.049 | 92.5 | 3.640 | 97.8 | 4.805 |
| 0.6 | oracle | 90.0 | 3.309 | 94.8 | 3.950 | 98.9 | 5.214 |
| | proposed | **89.2** | 3.258 | **94.3** | 3.882 | **98.6** | 5.102 |
| | naive | 100.0 | 12.746 | 100.0 | 15.215 | 100.0 | 20.083 |
| | SBL-true$k$ | 89.9 | 3.322 | 94.9 | 3.966 | 98.9 | 5.235 |
| | SBL-BIC | 87.8 | 3.139 | 93.4 | 3.746 | 98.2 | 4.945 |

Table 20: Means and standard errors (in parentheses) of MSE ($\times 10^{-3}$) for $\beta_i$. The smallest value for each experimental configuration is bolded, excluding those from the oracle method.

| | $p = 40, n = 200, k = 10, SNR = 40$ | | |
|---|---|---|---|
| $\rho_X$ | 0.2 | 0.4 | 0.6 |
| oracle | 6.766 (0.076) | 8.542 (0.100) | 13.025 (0.159) |
| proposed | **7.897** (0.102) | **9.730** (0.123) | **14.468** (0.181) |
| naive | 48.612 (1.191) | 87.180 (2.198) | 209.389 (5.418) |
| SBL-true$k$ | 7.968 (0.100) | 9.796 (0.122) | 14.533 (0.198) |
| SBL-BIC | 9.019 (0.138) | 10.937 (0.160) | 15.664 (0.230) |

