# OpenReview forum: "Uncertainty Quantification in Linear Regression With Mismatched Data"
_TMLR — Rejected by TMLR_

### Review · Reviewer_5cai · 2025-06-10

**Summary Of Contributions:**

The authors present the mismatched data linear regression problem as a generalized fiducial inference (GFI) task.  They show that subject to constraints, the fiducial distribution concentrates asymptotically around the true model parameters. They present a practical algorithm for generating samples from this fiducial distribution. The method is finally evaluated with respect to confidence and prediction sets (coverage, interval lengths) on synthetic data, and with prediction intervals and mean squared prediction error on real data. The evaluation shows good performance against selected baselines.

**Audience:**

Yes

**Broader Impact Concerns:**

I do not think this paper warrants broader a deeper discussion about the ethics. The ethical implication in this work is not much different from the baseline methods, and the applications with ethical implications are quite narrow.

This said, I would have appreciated a discussion about the role of this type of methodology in triangulation of anonymized data, and the risk of revealing individuals with sensitive private information.

**Claims And Evidence:**

Yes

**Requested Changes:**

Critical:

1. Add the Chakraborty & Datta Bayesian method in the comparisons.

2. Explain the statistical properties of the derived GFI confidence intervals.

3. Discuss the practical use of Theorem 4.1. When are the assumptions expected to hold or not? Relate this discussion to the simulation and real world examples.


Would strengthen:

1. Motivate BIC

2. State clearly that $\Sigma$ below equation 4 would have to be a **known** covariance matrix, not subject to inference.

3. make the ‘winners’ in Table 2-7 and 9 bold similar to table 8.

4. Explore lower SNR in simulation study and relate it to real world SNR.

5. Relate the Real Data example to the assumptions in Theorem 4.1. Are there any reasons to believe this will hold?

6. Discuss implementation complexity, CPU and RAM requirements.

**Strengths And Weaknesses:**

*Strenghths*

I appreciate the brief presentation of the GFI framework, and the presentation of the problem task is quite clear. The algorithm description is clear. The concentration result Theorem 4.1 lends credibility to the work. The simulation study is quite nice, presenting the approximate confidence intervals and prediction intervals, which both are of relevance. I laud your transparency about ‘No single method consistently outperforms’. I also think the real world data example provides value.

*Weaknesses*

I am confused why the work of “Chakraborty & Datta (2024)” is cited in the introduction, but not used as a bench mark. It is a natural comparison, given the repeated parallels between Bayesian and GFI methods. Generally, the connection to Bayesiansm could be expanded a bit in the paper; it is my understanding that the fiducial “confidence intervals” are more akin to credible intervals, and that the coverage guarantees are not provided in the sense of frequentist confidence intervals $\mathcal C$ for  parameter $\theta$, at level $\alpha$: $P[\theta \in \mathcal C] >= 1-\alpha$ . Not stating what the statistical properties of the confidence intervals are makes me as a reader unnecessarily suspicious.

The GFI framework allows several design freedoms. Most notably, you throw in the BIC regularization withour further discussion. While I suppose that it is chosen for giving being tractable results, this should be commented on in some way. Did you try other regularizers? Why/why not?

In Theorem 4.1, the requirement on $\tilde k \log \frac{n}{\tilde k} = o(n)$ seems quite strong. It rules out the situation where $\tilde k \propto n$, which is the setting in the simulation study, confusingly. One sufficient condition would be $\tilde k / n \to 0$, but I feel this assumption should trivially give a concentration result, since the fraction of mismatched samples would be negligable. Just subsample the dataset randomly, and the probability that there are mismatched data in the subset goes to zero. We would trivially get convergence in probability to the correct parameters (beta, sigma). In light of this naive asymptotics, can you identify situations where Theorem 4.1 provide useful conclusions more powerful than this naive approach?

Parameters in the simulation study is not that well motivated. The SNR 20-80 seems quite high. I looked into the El Nino dataset, which seems to be SNR 10 roughly. So why only look into this high SNR range for the simulation study? In the real world data example, the difference between methods in performance is small. A harder setting, where the methods separate more in performance would be good. If the methods really are that close in performance, I think the discussion should reflect this, and instead discuss complexity of implementation, CPU/RAM requirements, ease of integration in other frameworks, etc.

Finally, a question rather than a strength/weakness. Why does all methods in table 2 show empirical coverage below the nominal? This makes it seem like no method is suitable for the inference task at hand.

---

> ### Author Response · Authors · 2025-07-29
> **Response to Reviewer 5cai (1/2)**
>
> We thank Reviewer 5cai for the constructive suggestions and insightful comments regarding the simulation settings and the practical relevance of the assumptions used. We have carefully addressed these points in both the revised manuscript and our detailed responses below. For easy referencing, new or modified material in the revision is highlighted in blue.
>
>
> *Q1 :Add the Chakraborty \& Datta Bayesian method in the comparisons.*
>
>
> We appreciate the reviewer’s suggestion to discuss the comparison with Chakraborty \& Datta (2024). These authors build a fully Bayesian model using the joint posterior. The sampling method is the Hamiltonian Monte Carlo algorithm implemented in probabilistic
> language Stan. This method is not user-friendly for general practitioners unfamiliar with specialized Bayesian tools (Stan). Furthermore, this paper does not report the simulation result for empirical coverage of confidence intervals for model parameters (e.g. $\beta_i$),  which limits direct comparison. In contrast, our method is implemented using widely available Python libraries and provides a fast solution for generating fiducial samples of the model parameters to construct confidence intervals. We have contacted the authors to request their code, but unfortunately, we have not received a response from them.
>
>
> *Q2: Explain the statistical properties of the derived GFI confidence intervals.*
>
>
> The GFI confidence intervals in our work exhibit statistical properties similar to those of traditional frequentist confidence intervals. Under regularity conditions, they achieve nominal frequentist coverage probabilities asymptotically (Corollary 3.1 [1]). For example, in Table 2 under SNR=40, $92.2$% of confidence intervals cover the parameter $\beta_i$, which is close to the prespecified confidence level $95$%.
>
>
> *Q3: Discuss the practical use of Theorem 4.1. When are the assumptions expected to hold or not? Relate this discussion to the simulation and real world examples. Besides, the requirement on $\tilde{k} \log \frac{n}{\tilde{k}} =o(n)$ seems quite strong.*
>
> The assumptions in Theorem 4.1 impose a lower bound on the signal-to-noise ratio (SNR), which is order of $k^3\log \frac{n}{k}$ (here we assume $\tilde{k}/k$ is a constant). The requirement on $\tilde{k}$ is necessary to ensure consistent estimation.
> For example, Equation (15) in Slawski \& Ben-David (2019) shows that $||\tilde \beta-\beta^* ||_2$ can diverge if $\tilde{k}/n\to C_1$, where $0<C_1<1$.
>
> In our simulation settings, these assumptions are satisfied in most configurations. However, when the SNR is low, the mismatch ratio is high(e.g., $k/n=0.08$), or the sample size is small (e.g., $n=100$), the SNR condition does not hold with small $\delta$ (it reflects a nonzero probability of failing to recover the true permutation matrix). In such cases, a slight degradation in coverage rate of confidence intervals is observed. Nevertheless, the proposed method usually has the smaller MSE compared to other approaches.
> A similar trend is observed in the real data analysis. Based on our estimates, the SNR in the real dataset is approximately 10, which helps explain why the empirical coverage rates are slightly lower than those reported in Table 3 under higher SNR conditions. Also note that the real data setting differs from the simulations in dimensions,  $n=300, p=4$ in real data and $n=200, p=40$ in simulations.
>
>
> *Q4: Motivate BIC*
>
>
> The previous studies within GFI shows the promising result in model selection using BIC([1][2]). Besides, the simulation results confirm the performance.
>
>
> *Q5: State clearly that $\Sigma$ below equation 4 would have to be a known covariance matrix, not subject to inference.*
>
>
>
> It can be unknown, such as real data. It can be estimated using the sample covariance matrix, after which the input $X$ is standardized accordingly. Since the focus of this paper is on inference under mismatched data rather than covariance estimation, we do not further pursue inference for $\Sigma$ in this work.
>
>
>
> *Q6: make the ‘winners’ in Table 2-7 and 9 bold similar to table 8.*
>
>
> That is a good idea.  To ensure a fair comparison, best results are bolded (other than oracle and SBL-true$k$ methods). Please check the revised version.
>
>
> *Q7: Explore lower SNR in simulation study and relate it to real world SNR.*
>
> We have extended our simulation study to include lower SNR scenarios with SNR = $5,10,15$. These additional results are presented in Table 10 and 11 as complementary tables to Tables 2 and 3.
> As expected, the coverage rate of the proposed method degrades in low-SNR settings.
> This outcome aligns with the SNR conditions in Theorem 4.1. For the real data application, the estimated SNR is approximately 10. As discussed in our response to Q3,  this SNR level explains the slightly reduced empirical coverage observed in the real data.

---

> > ### Author Response · Authors · 2025-07-29
> > **Response to Reviewer 5cai (2/2)**
> >
> > *Q8: Relate the Real Data example to the assumptions in Theorem 4.1. Are there any reasons to believe this will hold?*
> >
> > Please refer to the reply to Q3.
> >
> >
> > *Q9: Discuss implementation complexity, CPU and RAM requirements.*
> >
> >
> > The algorithm is straightforward to implement in Python using common libraries, including NumPy, SciPy, cvxpy (optimization) and scikit-learn (kernel PCA). All experiments were conducted on a laptop equipped with CPU  Intel i7-12700H and RAM 16GB. Each repetition of the simulations takes approximately 1 second.
> >
> >
> > *Q10: Why does all methods in table 2 show empirical coverage below the nominal? This makes it seem like no method is suitable for the inference task at hand.*
> >
> >
> > This result can be explained by Theorem 4.1. Specifically, the probability bound below Equation (15) includes the constant $\delta$, which reflects a nonzero probability of failing to recover the true permutation matrix. When such errors occur, the resulting estimates may be systematically off-target, which in turn lowers the empirical coverage of the confidence intervals.
> >
> > Even when the signal-to-noise ratio (SNR) is relatively high, there remains a small probability (governed by $\delta$) that the selected permutation deviates from the truth. This can cause the constructed intervals to miss the target parameter. Importantly, this limitation is not unique to our method, but arises from the inherent difficulty of the permutation recovery problem.
> >
> >
> > Reference:
> >
> > [1] Lai, R. C., Hannig, J., \& Lee, T. C. (2015). Generalized fiducial inference for ultrahigh-dimensional regression. Journal of the American Statistical Association, 110(510), 760-772.
> >
> > [2] Hannig, J., Iyer, H., Lai, R. C., \& Lee, T. C. (2016). Generalized fiducial inference: A review and new results. Journal of the American Statistical Association, 111(515), 1346-1361.

---

### Review · Reviewer_vVfG · 2025-06-22

**Summary Of Contributions:**

This paper introduces a novel method for uncertainty quantification in linear regression under data mismatch using the generalized fiducial inference framework. It constructs fiducial confidence intervals for the regression parameters and predictive intervals for the responses, establishes theoretical guarantees for the recovery of the correct permutation pattern, and illustrates the proposed methodology on synthetic data and the El Nino data set from the UCI repository.

**Audience:**

Yes

**Claims And Evidence:**

Yes

**Requested Changes:**

See Section **Weaknesses**, which encompasses
* expand numerical work;
* ideally establish distributional consistency of the model parameters.

**Strengths And Weaknesses:**

**strengths**

* The problem of linear regression under data mismatch has been studied in the literature, as documented in the paper’s literature. However, the issue of uncertainty quantification of the model parameters and predictions has not been addressed in prior work. This paper aims to fill this gap, leveraging the generalized fiducial inference framework.
* At the methodological level, under a Gaussian design, the paper leverages ideas in Lai et al. (2015) to obtain a fiducial distribution for the model parameters –regression coefficient, permutation matrix, variance parameter. At the algorithmic level, it adapts the procedure from Slawski and Ben-David (2019) for estimating the regression coefficient and the permutation matrix. Then, it proposes a practical procedure for generating fiducial samples for the model parameters that can be used for the construction of confidence intervals. A procedure for obtaining prediction intervals for new observations is also given.
* Under standard regularity conditions available in prior work the paper establishes that with high probability, the probability assigned under the proposed fiducial distribution to incorrect permutations (and hence models) asymptotically vanishes.

**weaknesses**
* The key theoretical result obtained in the paper, is reminiscent of selection consistency results in Bayesian sparse linear -- specifically, that with high probability, the posterior ratio between the true and estimated sparsity pattern asymptotically diverges. A natural question arises:  can the authors obtain consistency of the entire generalized fiducial distribution for the model parameters, perhaps under additional regularity conditions? For potential directions, see results in Hannig (2014)
* The empirical evaluation is currently limited to cases where the mismatch proportion $k/n$ is small (up to 0.08), whereas Slawski and Ben-David (2019) examined a broader range up to 0.5. It would be valuable to assess how the performance of the proposed method degrades (or holds) as $k/n$ increases.
Additionally, exploring the impact of varying the dimension $p$ --both smaller and larger than the current setting ($p$=40) -- would help gauge robustness across different dimensional regimes and better contextualize the results relative to prior work.
* Although from the theoretical viewpoint, a correlated design for the predictors does not make any difference, what is the impact in practice? Since this aspect has not been empirically explored in either this or prior work, a brief study or discussion on how predictor correlation influences estimation and uncertainty quantification would strengthen the paper.

---

> ### Author Response · Authors · 2025-07-29
> **Response to Reviewer vVfG**
>
> We thank Reviewer vVfG for the constructive suggestions and insightful comments regarding the simulation settings and distributional consistency. We have carefully addressed these points in both the revised manuscript and our detailed responses below. For easy referencing, new or modified material in the revision is highlighted in blue.
>
>
>
> *Q1: expand numerical work;*
>
> We have extended our simulation study to include scenarios with $k/n=0.1,0.12,0.14$ and $p=10,20,40,60,80$. These additional results are presented in Tables 13-17 in Appendix C as complementary tables to Tables 4 and 5. As expected, we observe that the performance of the proposed method degrades when either  $k/n$ or $p/n$ becomes large, which is consistent with the assumption of SNR in Theorem 4.1. Notice that the constant $\delta$ appears in the condition of SNR and the probability bound (below Equation (15)). In practical terms, even when SNR is relatively high, there remains a small chance (governed by $\delta$) that the method selects an incorrect permutation, resulting in confidence intervals that fail to contain the true parameter.
> We also note that while some scenarios (e.g., Tables 10, 13 and 15) show that the naive method achieves empirical coverage closer to the nominal level, it does so at the cost of substantially wider confidence intervals.
>
> *Q2: ideally establish distributional consistency of the model parameters.*
>
>
>
> Theorem 4.1 shows the consistency of the generalized fiducial density in recovering the true permutation matrix, which is a key latent structure in the model. This recovery plays a central role in uncertainty quantification for the model parameters. As a result, the proposed method achieves asymptotically correct coverage of confidence intervals and consistent point estimation (Corollary 3.1 [1]).
>
>
> *Q3: Although from the theoretical viewpoint, a correlated design for the predictors does not make any difference, what is the impact in practice?  Since this aspect has not been empirically explored in either this or prior work, a brief study or discussion on how predictor correlation influences estimation and uncertainty quantification would strengthen the paper.*
>
> We have added simulations for the correlated design. Please see Tables 18–20 in Appendix C. The results confirm that the correlated design slightly improves the empirical coverage rate.
> This is consistent with redefining the regression coefficient as  $\tilde{\beta} = \Sigma^{1/2} \beta$  (see the explanation below Equation (4)). Using the same simulation setting with $n = 200$, $p = 40$, $SNR = 40$, and $\rho_X = 0.6$, we obtain  $|| \tilde{\beta} ||_2^2 / || \beta ||_2^2 = 3.8125$,  indicating a substantial amplification of signal. This helps explain the improved coverage performance.
>
>
> Reference:
>
> [1] Lai, R. C., Hannig, J., \& Lee, T. C. (2015). Generalized fiducial inference for ultrahigh-dimensional regression. Journal of the American Statistical Association, 110(510), 760-772.

---

### Review · Reviewer_4ihf · 2025-07-15

**Summary Of Contributions:**

The paper presents a principled approach for uncertainty quantification in
linear regression when data mismatch occurs. Building on the generalized
fiducial inference framework, the authors introduce a method for generating
fiducial samples, which allows for construction of confidence and prediction
intervals. Theoretical properties of the proposed method are established, and
numerical studies on simulated and real data are performed to evaluate the effectiveness of the algorithm.

**Audience:**

Yes

**Claims And Evidence:**

No

**Requested Changes:**

Address the problems pointed out in Weakness.

**Strengths And Weaknesses:**

### Strengths:

The paper addresses a gap in the literature by focusing on uncertainty
quantification in linear regression with mismatched data, an area that has been
largely overlooked in previous work.

### Weakness:

1. The presentation quality of the paper is lacking, especially from Section 2
   onward. There are numerous undefined notations and gaps in the exposition,
   making it difficult to follow the details of the proposed method. For
   example, the meaning of "fiducial density" is unclear; it is not specified
   whether the $G$ function in (6) is known or unknown, and it is ambiguous
   whether the expressions in (7) and (8) are general or specific to the linear
   model considered in this paper. Additionally, some notations are defined
   without being used, which increases the cognitive load for the reader. For
   instance, $\omega$ and $\Theta$ are introduced in Section 1.4 without
   explanation and are never used in other places of the paper.

2. The motivation and rationale for adopting certain techniques are not clearly
   discussed. For example, it is not explained why the generalized fiducial
   inference (GFI) approach is preferred over alternatives, or why the BIC is
   incorporated on page 5.

3. Theoretical results are not presented rigorously. For example, in Theorem
   4.1, the result in (15) is asymptotic (as $n \to \infty$), but it is then
   stated that this holds with a probability bound involving $n$, making it
   hard to interpret the result.

---

> ### Author Response · Authors · 2025-07-29
> **Response to Reviewer 4ihf**
>
> We thank Reviewer 4ihf for the constructive suggestions and insightful comments regarding the representation and motivation. We have carefully addressed these points in both the revised manuscript and our detailed responses below. For easy referencing, new or modified material in the revision is highlighted in blue.
>
>
> *Q1: The presentation quality of the paper is lacking, especially from Section 2 onward. There are numerous undefined notations and gaps in the exposition, making it difficult to follow the details of the proposed method...*
>
>
> We appreciate the reviewer’s feedback and acknowledge that the presentation in the original manuscript required improvement for clarity and completeness. In the revised version, we have thoroughly revised the exposition to enhance readability and ensure that all notations are properly defined when first introduced.
>
> Regarding the the reviewer’s specific comment: the function $G$ is a known function which defines the model equation and in our setting, it is explicitly given in Equation (9). The expressions in Equations (7) and (8) are general formulations intended to introduce the background of GFI; they are not restricted to linear models.
>
>
> *Q2: The motivation and rationale for adopting certain techniques are not clearly discussed. For example, it is not explained why the generalized fiducial inference (GFI) approach is preferred over alternatives, or why the BIC is incorporated on page 5.*
>
>
>
> GFI has been successful for other inference problems ([1][2][3]), so we expect it would work well for this problem too.  We highlight that GFI is not the only approach for uncertainty quantification for this problem - researchers can certainly try other approaches, but it is beyond the scope of this paper.
> Similarly, the previous studies within GFI shows the promising result in model selection using BIC([1][2]).
>
>
> *Q3: Theoretical results are not presented rigorously. For example, in Theorem 4.1, the result in (15) is asymptotic (as $n \to \infty$), but it is then stated that this holds with a probability bound involving $n$, making it hard to interpret the result.*
>
>
> Thanks for pointing out and we have rewritten it. Please check the revised version.
>
>
> Reference:
>
> [1] Lai, R. C., Hannig, J., \& Lee, T. C. (2015). Generalized fiducial inference for ultrahigh-dimensional regression. Journal of the American Statistical Association, 110(510), 760-772.
>
> [2] Hannig, J., Iyer, H., Lai, R. C., \& Lee, T. C. (2016). Generalized fiducial inference: A review and new results. Journal of the American Statistical Association, 111(515), 1346-1361.
>
> [3] Wei, Z., \& Lee, T. C. (2023). High-dimensional multi-task learning using multivariate regression and generalized fiducial inference. Journal of Computational and Graphical Statistics, 32(1), 226-240.

---

### Decision · Action_Editor_YFA9 · 2025-09-09

**Recommendation:** Reject

**Additional Comments:**

In addition to clarity issues raised by the reviewers, a resubmission should address the two points raised above.

### **Presenting the limitations of the method**

I have a number of recommendations to make these more transparent.

1. Add a limitations section e.g. to Conclusion.
2. Qualify claims in the abstract rather than claiming to "demonstrate its practical effectiveness".
3. Add plots to the results section that show the effectiveness of the method as function of key experimental covariates, including $p$, SNR and $k/n$.

### **Defining the "best" results**

4. Define "best" results transparently. As a starting point, I would suggest a definition that the best method is the one whose coverage is greater than or equal to nominal and that has the shortest interval among such methods.

**Audience:**

Yes

**Audience Explanation:**

All reviewers agree that some readers would be interested in the paper.

**Claims And Evidence:**

No

**Claims Explanation:**

The submission makes multiple claims that are not supported by sufficient evidence.

1. It presents the proposed algorithm as a general solution to uncertainty quantification in mismatched linear regression e.g. in abstract and contributions without properly discussing its limitations. The additional experiments requested by the reviewers to explore the limitations are reported as tables in supplementary material without any discussion of their results in the main text.
2. Result tables mark "best result" in bold that is always the proposed method without defining what "best" means.

Suggestions for addressing these below in additional comments.

I believe that addressing these will require another round of reviews and therefore the submission should be rejected with possibility of resubmission of a major revision.

**Resubmission Of Major Revision:**

The authors may consider submitting a major revision at a later time.